# Spatial layout characteristics of Northern Wei Luoyang: A spatial humanities analysis of the "Record of the Monasteries of Luoyang"

**Runfeng Sun, Wenjia Liu**◯*

School of architecture, Zhengzhou University, Zhengzhou, Henan, China

* sandylwj@zzu.edu.cn

## Abstract

This paper explores the urban spatial structure and social stratification of Northern Wei Dynasty Luoyang City, based on *Record of the Monasteries of Luoyang*. Adopting a spatial humanities approach, the study integrates quantitative approaches, including the application of geographic information systems (GIS) for spatial analysis and complex network analysis. The findings indicate that the spatial organization of Northern Wei Luoyang was primarily concentrated within and around the inner city. People of different social classes and statuses occupied distinct spaces, forming socially stratified groups. As one moves outward from the city's center, the social rank of these groups decreases. The city's layout is characterized by a single central axis, with multiple spatial divisions, and the west side being associated with higher-status groups. This spatial arrangement was shaped by factors such as geography, urban development, Central Plains rituals, traditional customs, and ethnic diversity. This study enhances our understanding of the spatial layout characteristics of Northern Wei Luoyang. It provides innovative insights and methods for understanding the city's structure and its implications for other underground historical cities. Furthermore, it provides valuable data and visualizations that could support future preservation and planning efforts for the city.

## Introduction

Historically, humanities research has primarily focused on temporal and qualitative aspects, often overlooking the potential of spatial concepts and quantitative methods [1]. However, advancements in computer science and the rise of positivist methodologies since the twentieth century have facilitated the adoption of digital research techniques. For example, Geographic Information Systems (GIS) have been utilized in archaeology and other humanities disciplines for over forty years. Yet, only in recent years have further technological advancements led to the emergence of what is now increasingly recognized as spatial humanities. This evolving field harnesses

**Data availability statement:** All relevant research data have been organized and publicly archived in protocols.io, including all relevant information and detailed annotations. The data can be accessed via the following DOI: 10.17504/protocols.io.rm7vzk1n4vx1/v1.

**Funding:** The author(s) received no specific funding for this work.

**Competing interests:** The authors have declared that no competing interests exist.

advanced tools to explore the spatial dimensions of human culture and history in greater depth [2].

Technologies like GIS, known for their robust analytical capabilities, equip researchers to uncover novel issues and perspectives, enhancing the breadth and depth of humanities research [3,4].Initially, GIS applications in spatial humanities primarily focused on mapping national or regional historical boundaries and geographic demarcations [5,6]. Over time, as GIS methodologies advanced within the field, there has been a significant shift towards more human-centric studies, with greater focus on humanity and applications like heritage preservation.[7–12]. The integration of various digital technologies in humanities has elevated the complex network approach as a novel method within spatial humanities research. Scholars such as Barabasi [13], Watts [14], and others have refined complex network models, characterizing them by their small-world and scale-free properties. By merging this approach with GIS and geospatial data, researchers can effectively capture and represent the inherent complexities of geographic, urban, ecological, and geophysical systems [15]. This method is now increasingly applied in spatial humanities to enrich studies in history [16], literature [17], religion [18], and other fields, demonstrating its broad utility and relevance. Today, spatial humanities has expanded to incorporate various technologies and methods, including corpus linguistics, spatial hypertexts, mapping, and machine learning, to analyze cultural and social phenomena found in historical texts, novels, manuscripts, and ancient [19–24]. Currently, there are relatively few studies that explore ancient Chinese underground historical cities and their social patterns as recorded in ancient texts that combine both literary and documentary elements.

The Northern Wei Luoyang City, dating from the Northern Wei Dynasty (386–534 AD), represents a critical phase in the evolution of the Han-Wei Luoyang urban landscape. The origins of Han-Wei Luoyang City trace back to the Western Zhou Dynasty (11th century BCE). Significant construction occurred during the Eastern Han Dynasty (1st century CE), followed by further development under successive dynasties. Over time, the city's design evolved from a multi-palace system, where the emperor's position was decentralized, to a single-palace system with the emperor positioned at the center of a unified axial layout. After the Northern Wei Dynasty, the city was eventually abandoned, with much of its ruins buried beneath villages and farmland [25,26]. Today, only the core area has been excavated and is on display (S1 File). During the Northern Wei period, Luoyang largely retained the fundamental layout and structure of its predecessors. However, new developments included the establishment of an outer city, creating a mature spatial organization divided into three levels: the palace, the inner city, and the outer city. This hierarchical design system marked a significant milestone in urban planning. (Fig 1)This layout marked the first instance of a Chinese capital city incorporating three distinct urban levels, Northern Wei Luoyang is regarded as the first major capital in ancient China to implement this tripartite spatial structure, influencing the design of later cities in China and East Asia [27,28]. In 1961, Northern Wei Luoyang was designated as one of the initial group of national key cultural relics protection units in China. The city's design, including its overall form, functional zoning, and planning methodologies, has had a

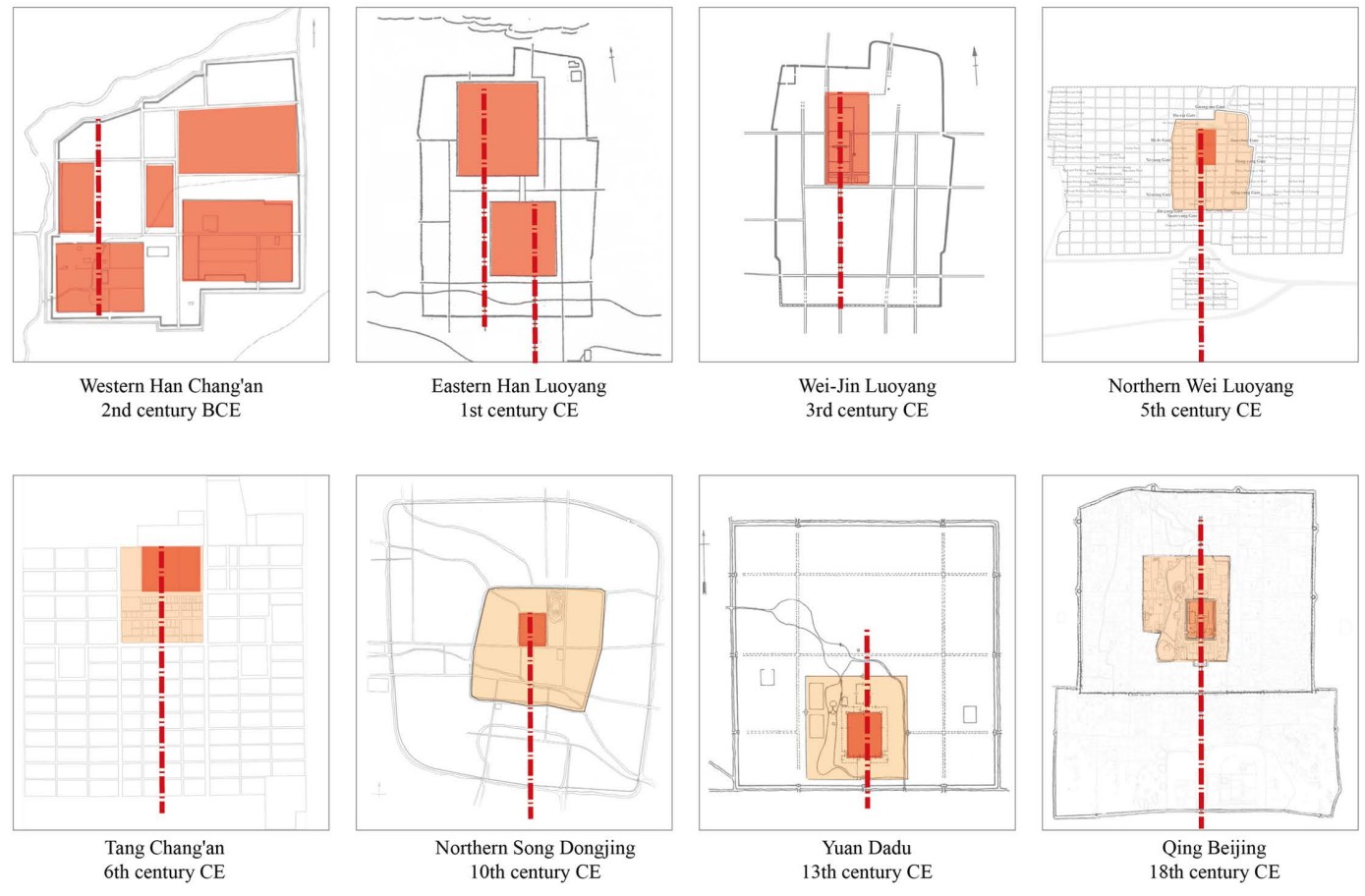

Western Han Chang'an
2nd century BCE

Eastern Han Luoyang
1st century CE

Wei-Jin Luoyang
3rd century CE

Northern Wei Luoyang
5th century CE

Tang Chang'an
6th century CE

Northern Song Dongjing
10th century CE

Yuan Dadu
13th century CE

Qing Beijing
18th century CE

**Fig 1. Diagram of Historical Chinese Cities.**

profound influence on the development of later Chinese capital cities and the architectural evolution of Chinese palatial systems [29,30].

Record of the Monasteries of Luoyang"洛阳伽蓝记", authored in 547 AD by Yang Xuanzhi (杨衒之,?-555), a minister of the Northern Wei Dynasty who lived during the Northern Wei period, is a seminal text in the historiography and spatial analysis of ancient Chinese cities. The document meticulously details the location, size, and architecture of 44 temples in Northern Wei Luoyang, alongside narratives of significant events associated with these temples. It provides a comprehensive depiction of the urban and social landscape of Luoyang during this era, making it an invaluable resource for understanding the city's historical and geographical context. Its detailed accounts are crucial for scholars studying the urban development of the Northern Wei Dynasty [31]. In this paper, we adopt the perspective of spatial humanities, integrating quantitative approaches such as GIS for spatial analysis and complex networks analysis to extract the spatial information embedded in these textual records. Analyzing the physical space and the society embodied in the events recorded in ancient canonical texts in order to uncover the parts that may have been neglected in the past studies, providing a new method for analyzing the spatial patterns of Chinese underground historical cities. Our goal is to uncover aspects that may have been overlooked in previous studies. Specifically, we systematize and analyze the functional and event spaces of Northern Wei Luoyang, as documented in the Record of the Monasteries of Luoyang, to reveal the spatial layout characteristics and social differentiation within the city. This study aims to refine and expand upon the existing understanding of Northern Wei Luoyang's urban space and society.

## Methods

### Scope of the Study and Subjects

The spatial focus of this study encompasses the Northern Wei Dynasty's city of Luoyang and selected surrounding areas. This includes the inner city, the outer city, the southern outskirts near the Four Barbarians' Lodging House (四夷馆), and Four Barbarians' Wards (四夷里). Temporally, the study covers the period documented in the Record of the Monasteries of Luoyang, beginning with the capital's relocation to Luoyang in 494 CE. and concluding with the division of the Northern Wei Dynasty into the Eastern and Western Wei Dynasties in 534 CE. This timeframe captures a critical forty year epoch in the city's history, providing a focused lens for examining the urban and social developments during the late Northern Wei period.

The research focuses of this paper are categorized into functional space and event space within the Northern Wei Dynasty's city of Luoyang, as depicted in the Record of the Monasteries of Luoyang. We employ spatial analysis techniques in GIS and complex network theory to visualize and analyze both the functional and event space layout features of the city. This approach allows for a detailed examination of the spatial distribution and interconnectivity of urban elements and historical events recorded during the era.

Functional space, a concept from urban geography, refers to a geographic unit within a city composed of various interconnected elements [32–34]. It represents the physical manifestation of urbanization processes and urban spatial structures [35]. In urban studies, the evolution of functional space closely mirrors changes in the urban spatial structure, reflecting both the development process and the shifting patterns of urban areas [36,37]. In this study, the functional space of Northern Wei Luoyang City is categorized into three primary types based on their distinct functions: monasteries, government offices and residential spaces. This classification helps to elucidate the organizational and functional dynamics of the city during the Northern Wei period.

Event space pertains to the locales of human activities, focusing on the processes and experiences that occur within these spaces. Defined as both architectural physical spaces and historical cultural contexts [38,39], event spaces are critical in understanding how social life influences urban environments [40]. The study of social interactions within historical cities offers deeper insights into urban spatial layouts and societal structures. The growing recognition among scholars of the importance of spatial attributes to events underscores this perspective [41–44]. In this study, the event space of Northern Wei Luoyang City is segmented into three categories based on the nature of the events: political activities, daily life, and auspicious legends. This classification aids in comprehensively analyzing how different types of events shape the spatial and social fabric of the city.

### Data sources

The primary textual source for this study is Fan Xiangyong's (范祥雍) "*Record of the Monasteries of Luoyang Annotation*" (洛阳伽蓝记校注) [45]. To address gaps in this text, supplemental sources have been utilized, enriching the historical context and details. These include *Henan Gazetteer*(河南志) [46], *Book of Wei*(魏书) [47], *Commentary on the Water Classic* (水经注) [48], *Comprehensive Mirror in Aid of Governance*(资治通鉴) [49] and History of the Northern Dynasties(北史) [50] among others. These documents collectively provide a comprehensive foundation for analyzing the historical and cultural landscape of the Northern Wei Luoyang City. This paper references the English version of the Record of the Monasteries of Luoyang, translated by Yi-t'ung Wang [51], for the English translation of terms from the original text.

The mapping of Luoyang during the Northern Wei Dynasty primarily relies on various historical and modern sources to create an accurate depiction. Key references include the "Record of the Monasteries of Luoyang annotation" "Record of the Monasteries of Luoyang Map"[45], "Henan Gazetteer" "Northern Wei Capital Map"[46](Fig 2) and "Jin-yong City Map"[46]. Additionally, contemporary restored maps, which have been validated by scholars, play a crucial role. Among these, the restoration map titled "Northern Wei Dynasty Luoyang Outer City Wards and Markets Pattern Speculation

Restoration Map" [52] by Qian Guoxiang is particularly noteworthy. This map, based on archaeological and historical data, represents one of the most comprehensive reconstructions of the city's layout.

Additionally, this study requires the collection of modern map data to accurately align historical maps with contemporary geographic references. This process involves standardizing the coordinate system to ensure consistency across different data sets. Modern road and water system data are sourced from open map platforms, including National Platform for Common GeoSpatial Information Services [53]. High definition image maps from platforms like Google Maps [54] and National Platform for Common GeoSpatial Information Services [53] are also employed to aid in the precise alignment of these maps. This methodological approach ensures a robust basis for comparing ancient and modern Luoyang, enhancing the accuracy of spatial analyses.

## Data processing

The original textual data was collected and verified using sources such as the *Record of the Monasteries of Luoyang Interpretation* and *Record of the Monasteries of Luoyang Annotation*, and subsequently entered into a database. The recorded text was then imported into MARKUS [55], an online text annotation platform for ancient books developed by Leiden University. MARKUS employs word processing algorithms and is linked to databases like the China Biographical Database (CBDB), which automatically extracts relevant information from the text, including locations, character names, official titles, and years (Fig 3). This allows for efficient annotation and data analysis.

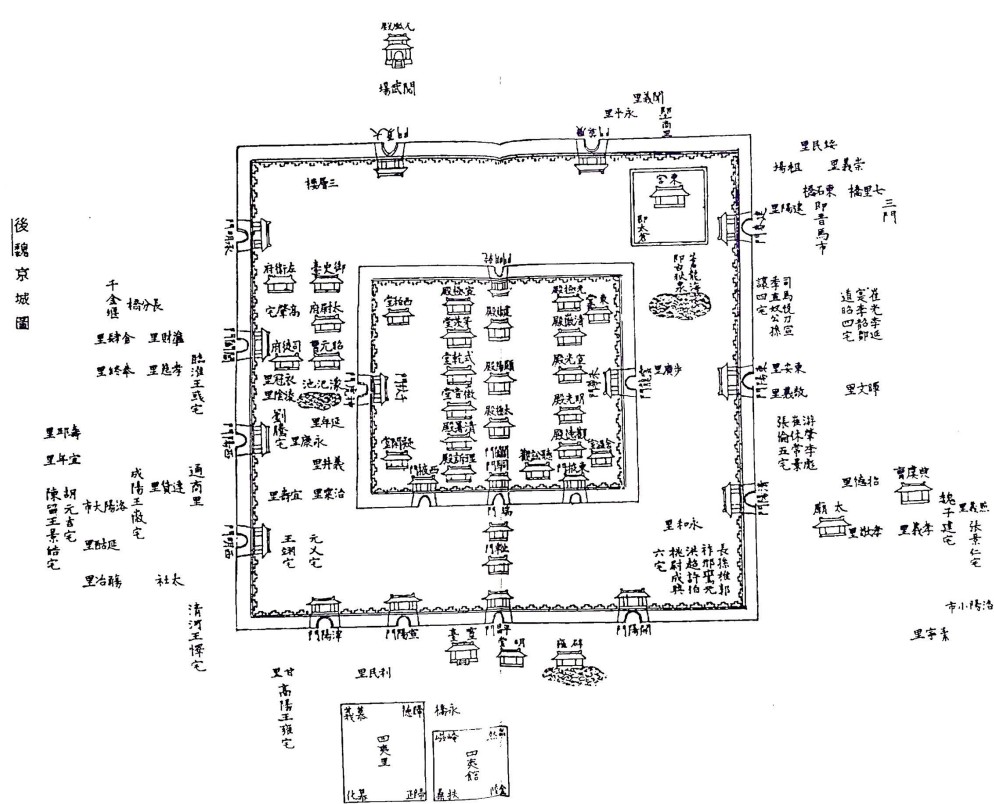

**Fig 2. Northern Wei Capital Map.**

Currently, the archaeological exploration of the gates and walls of Northern Wei Luoyang City has yielded comprehensive results, with extensive findings that have helped accurately determine the locations and names of these gates. These advancements have significantly enriched our understanding of the city's historical layout and infrastructure [56].

The Record of the Monasteries of Luoyang offer detailed descriptions of the locations of most temples and other spatial landmarks within the city, primarily in relation to prominent features such as city and palace gates. For instance, the record specifies, "The Yong-ning Monastery(永宁寺) was constructed in the first year of the Xi-ping period (Prosperous and Peaceful) (CE 516), by decree of Empress Dowager Ling, whose surname was Hu. It was located one li (Length unit, about 419m) south of the Chang-he Gate(閶闔門) on the west side of the Imperial Drive, facing the palace grounds." [45] These detailed accounts provide valuable insights into the urban geography of Northern Wei Luoyang, anchoring historical sites within the city's architectural context.

The geospatial positioning of Northern Wei Luoyang City has been meticulously refined by scholars, who have clarified the locations of numerous sites based on existing drawings. For sites whose locations remain uncertain, researchers have turned to the Record of the Monasteries of Luoyang and other historical documents to conduct analyses and establish their probable positions. This rigorous approach enables a more accurate reconstruction of the city's historical geography, enhancing our understanding of its spatial layout.

During the georeferentiation process, modern satellite images, survey maps, and maps depicting Luoyang during the Northern Wei Dynasty were integrated into GIS. To accurately determine the locations of key nodes within the ancient city, these maps were aligned with corresponding sites on modern maps. Notable landmarks such as Yong-ning Monastery, White Horse Temple(白马寺), and the Chang-he Gate, along with specific segments of the outer and inner city walls, were designated as primary control points. Additional points along the city walls served as auxiliary control points. These were crucial in making alignment corrections to ensure the precision of geographic coordinates on the historical maps. This meticulous method enhances the reliability of spatial data in reconstructing the ancient cityscape.

Following geographic alignment, the locations of the Northern Wei Luoyang's water systems, city walls, and residential wards were plotted in GIS, using the original text of the Record of the Monasteries of Luoyang along with detailed map analyses. This GIS-mapped base will serve as the foundation for further research. Building on this base map, both the functional spaces and event spaces within the city are meticulously mapped, facilitating a comprehensive spatial analysis in the subsequent stages of the study.

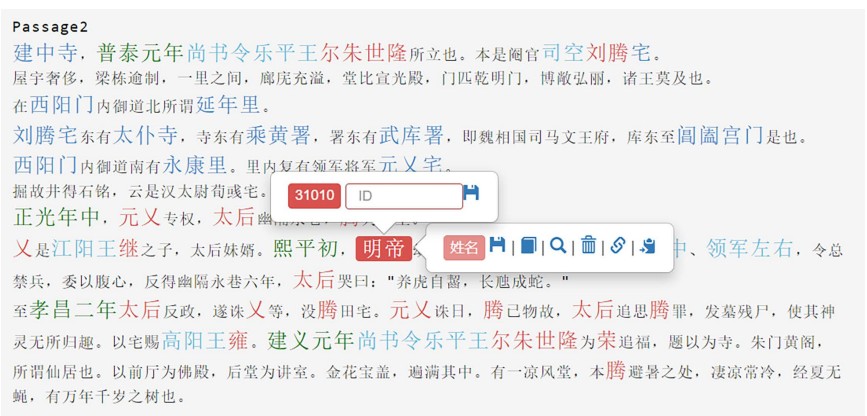

**Fig 3. Schematic diagram of MARKUS platform labeling.**

## Analysis of the functional spatial layout of the Northern Wei Dynasty Luoyang City in the Record of the Monasteries of Luoyang

In analyzing the functional spatial layout of Northern Wei Dynasty Luoyang, specific geographic locations from the Record of the Monasteries of Luoyang were paired with functional space nodes. These nodes were categorized into three types: monasteries, government offices, and residential spaces. Specifically, the city comprised 64 monasteries, 29 government offices, 51 residential spaces, totaling 144 functional spaces. These spaces were then mapped based on their geographic locations for subsequent analysis and visualization.

### Monasteries

**Analysis of the overall spatial layout of the monastery.** Monasteries were a prominent feature of the Northern Wei Dynasty's urban landscape in Luoyang, extensively documented in the Record of the Monasteries of Luoyang. The prevalence of Buddhism, supported ardently by the royal family and elite classes, conferred a unique social status upon the religion within the city [57,58]. The Record of the Monasteries of Luoyang specifies the locations of 64 monasteries, all of which were mapped and analyzed using GIS. This detailed mapping underscores the significant integration of Buddhism into the daily and spatial fabric of Luoyang, reflecting its esteemed position in society during this period.

Kernel density analysis, standard deviation ellipse analysis, and mean center analysis were conducted on the monasteries to produce spatial kernel density maps and spatial standard deviation ellipse maps (Fig 4).

The analysis reveals that monasteries were predominantly located near the inner city walls. Their distribution was primarily to the south, east, and west of the city center, with almost none in the northern sector, indicating a strategic placement near major transportation routes such as city gates and imperial roads. Notably, there was no significant monastic presence near the outer city's edge. Monasteries were centrally located within the inner city, often aligning with the city's central axis and near key gateways, emphasizing their integral role in the urban structure of Northern Wei Luoyang.

**Analysis of the spatial layout of different types of monasteries.** The Record of the Monasteries of Luoyang details that the construction of temples in Northern Wei Luoyang involved various social classes, predominantly Xianbei royalty, Han officials, and commoners. This documentation highlights the diverse contributors to the religious and architectural landscape of the city, reflecting the complex social hierarchy of the time.

Temples erected by the imperial family were strategically located at the heart of Northern Wei Luoyang, particularly along the vital inner city axis, underscoring their significance within the urban layout. Notably, the imperial family did not commission temples in the peripheral areas of the Four Barbarians' Lodging House and Four Barbarians' Wards, situated on the northern and southern outskirts of the city. This distribution highlights a clear demarcation in the religious and architectural priorities of the era.

Monasteries constructed by officials in Northern Wei Luoyang were predominantly positioned in subcentral locations within the city. These structures were mainly situated on the east side of the inner city, particularly near key access points such as outside the Jianchun Gate and near the Qingyang Gate, close to Xiaoyi Lane. This pattern indicates a general preference for locations near city gates, reflecting the strategic importance of these areas within the urban layout.

Monasteries built by commoners were primarily located in noncentral areas of Northern Wei Luoyang, often in the outer city and farther from the city gates. These monasteries were evenly distributed between the east and west sides of the city, displaying a symmetrical pattern. This symmetry suggests that Luoyang's central axis served not only as the dividing line for the outer city but also as the focal point for the spatial distribution across the entire city.

In comparing the three types of temple spaces, it is evident that the spatial distribution of temples built by different social classes was distinctly separated. This separation reflects a hierarchical spatial pattern, with higher classes occupying central locations and lower classes located further out. As noted by Eiji Sagawa, the spatial sequence along the Northern Wei central axis—from the Palace City to the outer sections—was divided into four tiers: the emperor, bureaucrats,

ordinary commoners, and foreign commoners. This hierarchy aligns with the spatial structure described in the Shangshu (尚书) [59]. The distribution of monasteries further supports this pattern, with the layout of Northern Wei Luoyang reflecting a clear classbased spatial hierarchy, decreasing from the inner to the outer city.

## Government offices

**Analysis of the overall spatial layout of the government offices.** From the relocation of the Northern Wei capital to Luoyang in the 18th year of Taihe (494 AD) until the dynasty's division in 534AD, Luoyang served as the capital for 40 years. During this period, the city housed not only its local administrative offices but also the central administrative organizations of the entire Northern Wei Dynasty. These official offices were integral components of Luoyang's functional space. The Record of the Monasteries of Luoyang explicitly mentions 29 official offices, which were entered into GIS for analysis, contributing to a deeper understanding of the city's administrative structure during this period.

Kernel density analysis, standard deviation ellipse analysis, and mean center analysis were conducted to create the spatial kernel density map and standard deviation ellipse map for the official offices in Northern Wei Luoyang (Fig 5).

The analysis reveals that government offices were predominantly located in the inner city, distributed along both sides of the central axis and near the Palace City on the east and west. Additional clusters appeared in the southeast and southwest corners of the inner city. Outside the city, there were few government offices, with notable exceptions in the east near Luoyang Prefecture, the Rent Field in the west, and Heyin Prefecture in the north, with almost no distribution

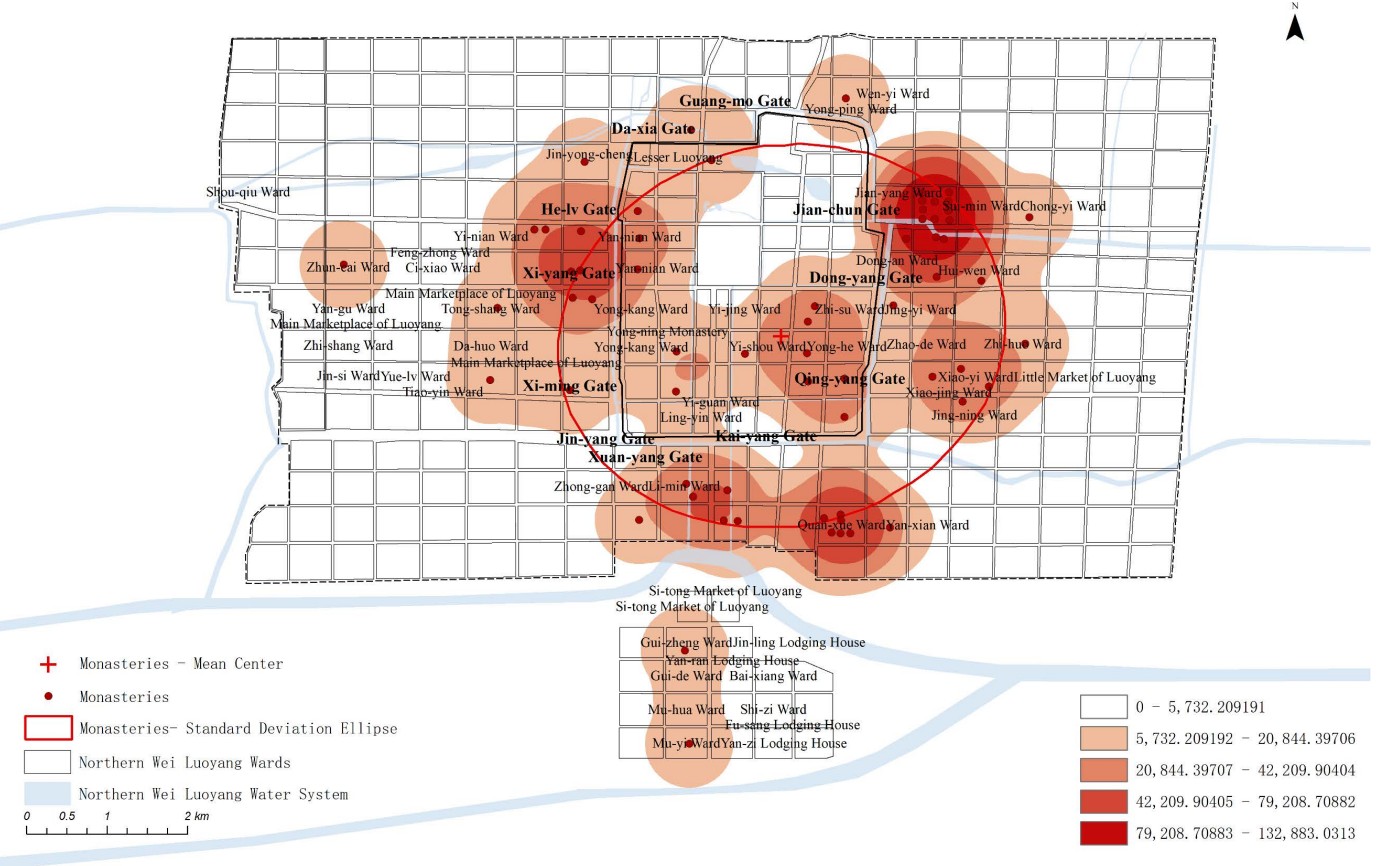

**Fig 4. Spatial GIS analysis map of the Monasteries.**

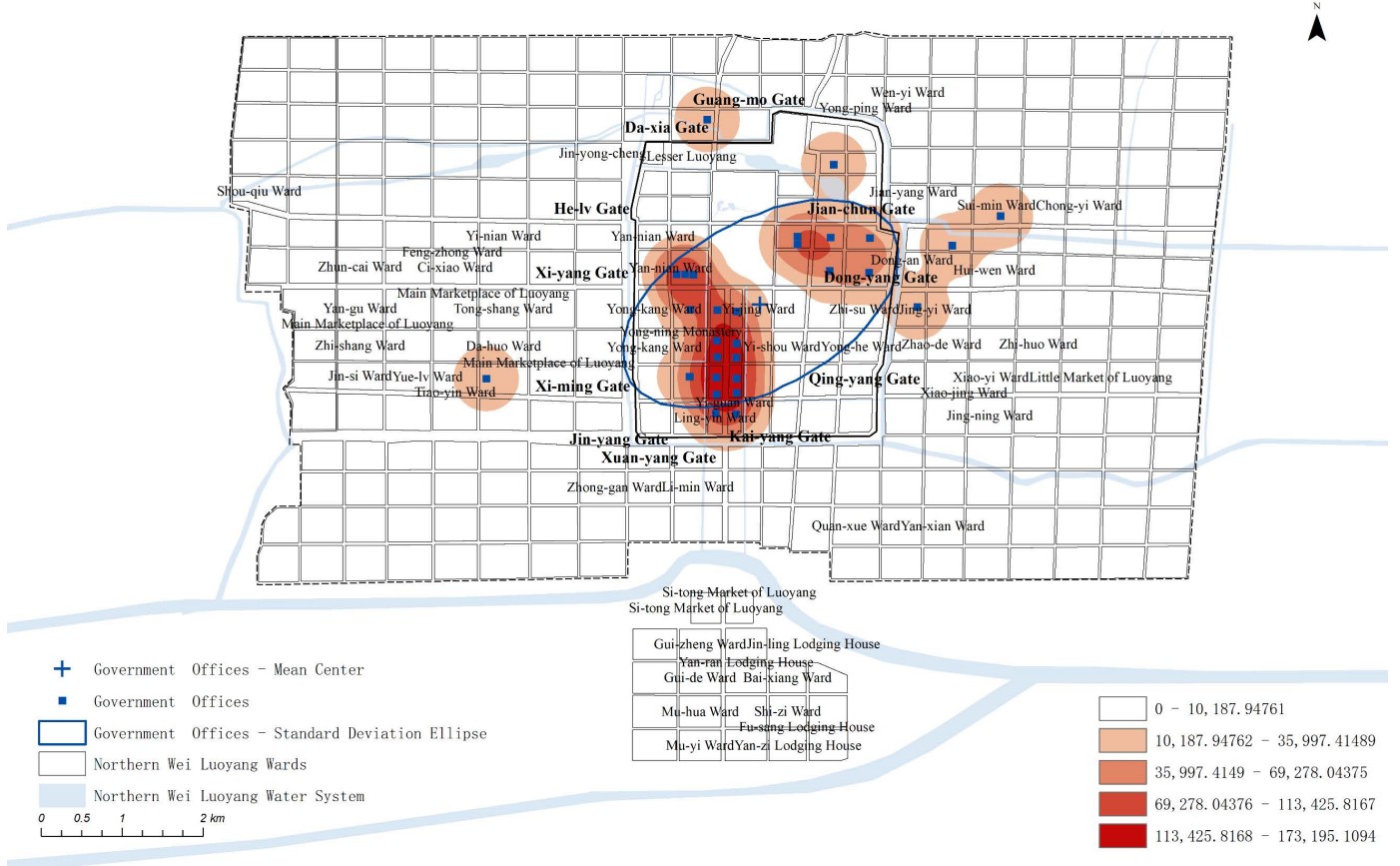

**Fig 5. Spatial GIS analysis map of the Government Offices.**

elsewhere. In addition to the Palace City, officials primarily occupied core urban spaces, such as both sides of the central axis and areas adjacent to the Palace City, forming an essential part of Northern Wei Luo-yang's ceremonial and administrative landscape.

**Analysis of the spatial layout of different types of government offices.** The official spaces responsible for various governance functions were distributed across different areas of the city. Offices holding state power were centrally located, while those managing food supply were situated strategically to serve their purpose. Additionally, grassroots governance organs were dispersed in various locations throughout the outer city, ensuring localized administration and governance across different regions of Luoyang. This spatial arrangement reflects the functional differentiation of official spaces in Northern Wei Luoyang, tailored to meet the specific needs of governance.

The official offices along the central axis, including the Office of the Grand Commandant (太尉府) and the Situ's Office (司徒府), served as key state administrative bodies in Northern Wei Luoyang. These offices oversaw functions related to administration, military, and education, symbolizing the central authority and power of the state.

South of the Palace City, near the Chang-he Gate on the west side, was the Court of the Imperial Stud(太仆寺), responsible for managing the emperor's carriages, horses, weapons, and other ceremonial functions, reflecting imperial authority. On the east side of the Palace City were institutions like the Imperial Granary(太仓署) and the office of the Imperial Palace

Parks(勾盾署), which managed food supplies. In the northeastern part of the inner city, was the office of He-nan Yin(河南尹), tasked with overseeing the entire capital. Although this office was located in a relatively remote part of the inner city, it was strategically close to the palace and the originally planned the palace of the crown prince.

The Luoyang and Heyin prefectures were situated to the east and west of the city, respectively. Luoyang Prefecture was located in Sui-min Ward(绥民里), while Heyin Prefecture was positioned southeast of the Main Marketplace of Luoyang. Both prefectures were situated near the more prosperous and densely populated areas of the outer city, highlighting their significance in managing these bustling regions.

Overall, the central axis of Luoyang was flanked by key state administrative institutions of the Northern Wei Dynasty. The western side of the axis housed official offices and spaces related to the imperial family and religious functions, while the eastern side contained offices responsible for food production. Meanwhile, the outer areas of the inner city were dedicated to grassroots administration, overseeing the governance of the outer city. This spatial arrangement highlights the hierarchical or-ganization of administrative functions within the city.

The layout of official offices in the inner city of Northern Wei Luoyang was shaped by the dynasty's ritual system, which positioned state power institutions along both sides of the central axis. However, spatial constraints in the original city design led to the eastern side of the Palace City being primarily occupied by food industry offices. Additionally, the governance limitations in the outer city necessitated the establishment of county offices within this area to manage administrative functions effectively.

**Residential spaces**

**Analysis of the overall spatial layout of the residential spaces.** The Record of the Monasteries of Luoyang primarily records the residences of the royal family and officials in various residential wards, with fewer references to the residence of the general population. Notably, some entries describe properties repurposed as temples. The Record of the Monasteries of Luoyang explicitly mentions the locations of 52 residential spaces, which have been entered into GIS for further analysis.

Kernel density analysis, standard deviation ellipse analysis, and mean center analysis were conducted to generate the spatial kernel density distribution map and the spatial standard deviation ellipse map for the residential spaces in Northern Wei Luoyang (Fig 6).

The analysis reveals that residential spaces were primarily concentrated in the west near Xi-yang Gate and on the east side, both away from and adjacent to the palace. In the outer city, residential areas were more densely distributed on the east and west sides, with fewer residential spaces in the north and south. As the main living areas for residents, these residential spaces occupied significant space outside the core areas, playing an important role in the overall spatial layout of Northern Wei Luoyang.

**Analysis of the spatial layout of different types of residential spaces.** In Northern Wei Luoyang, residential spaces were primarily organized based on class and social status. Residents of different social strata tended to cluster in distinct areas, forming house groups that reflected their status within the city. This spatial segregation highlights the class-based organization of residential areas in Luoyang during this period.

On the east side of the inner city, the residential wards between Dong-yang Gate(东阳门) and Qing-yang Gate(青阳门) were predominantly inhabited by officials. For example, in the nearby Zhi-su Ward(治粟里) area, officials who managed food supplies lived with their families. This part of the inner city was also a hub for temple spaces, where religious and residential spaces coexisted. The practice of donating residential spaces to be converted into temples significantly influenced the transformation of urban spaces in Luoyang, blending family dwellings with religious institutions.

In the eastern part of the city, residential wards such as Zhao-de Ward(昭德里) and Dong-an Ward(东安里) were primarily occupied by officials' residences. Additionally, the residential spaces of some officials and commoners were scattered throughout areas like the Little Market of Luoyang(洛阳小市) and Jian-yang Ward. The Little Market of Luoyang

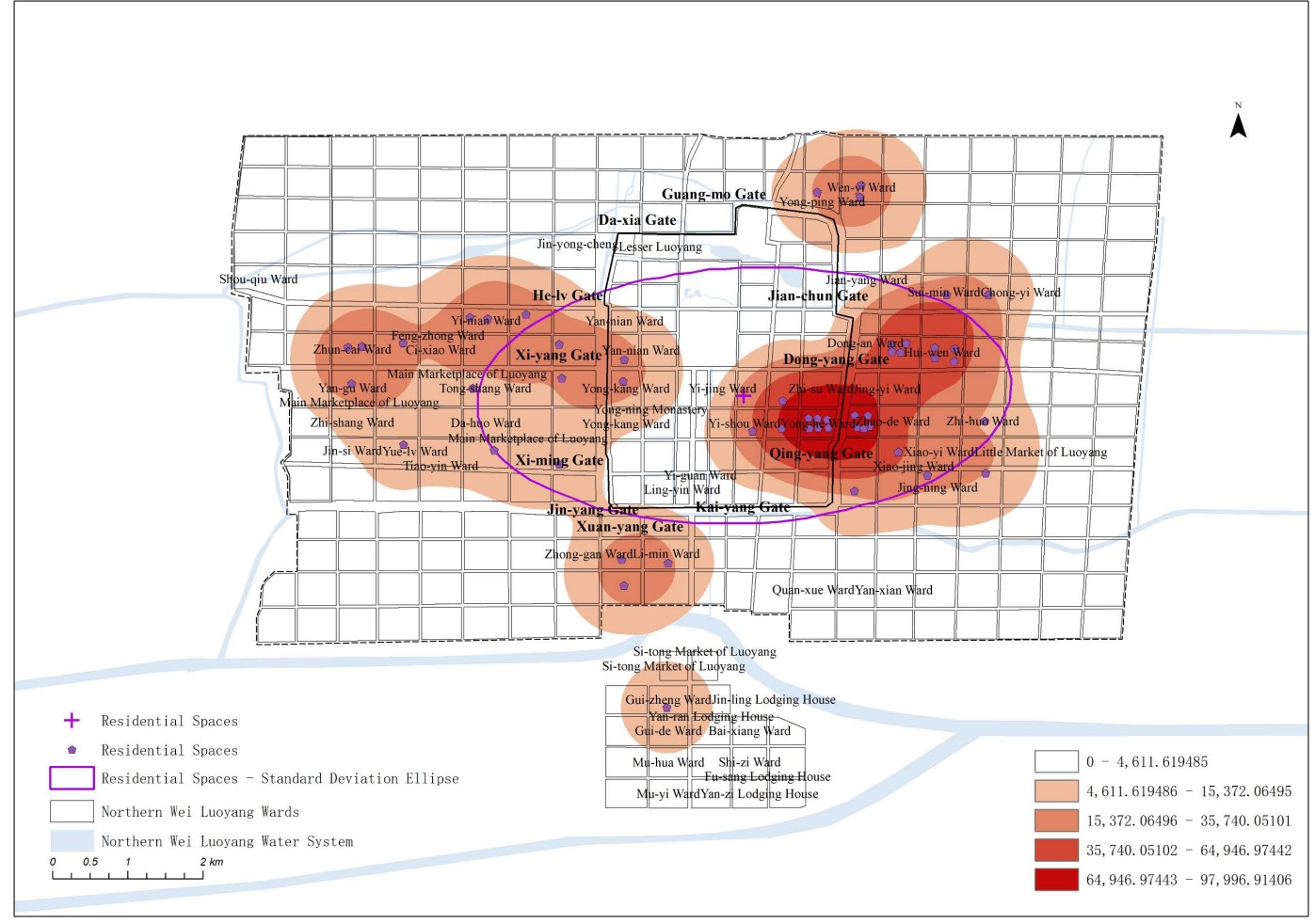

**Fig 6. Spatial GIS analysis map of the Residential Spaces.**

served as the commercial hub of the eastern city, while Jian-yang Ward was also a prosperous neighborhood. In contrast, the southern and northern parts of the city contained only a few scattered residential spaces.

On the west side, the area around the Main Marketplace of Luoyang was a commercial center, inhabited by people engaged in trade, handicrafts, and services. Notable areas included Yan-gu Ward(延酤里) and Zhi-shang Ward(治觞里), where residents were involved in the brewing industry. As the outer city extended closer to the inner city, the social status of homeowners increased, reflecting a spatial hierarchy based on class.

Additionally, the Record of the Monasteries of Luoyang mentions areas, such as Shou-qiu Ward(寿丘里), Jian-yang Ward, Four Barbarians' Lodging House, and Four Barbarians' Wards in the southern outskirts of the city, where large numbers of residential spaces were owned by various landlords, though detailed information about these properties is not provided. These areas further illustrate the spatial distribution of different residential spaces and social classes within Luoyang, highlighting the city's class-based residential pat-terns.

Shou-qiu Ward, consisting of seventeen residential wards on the western edge of the city, was t exclusively inhabited by members of the imperial family. The residents, all royal family members, competed with one another in constructing elaborate gardens and landscapes, reflecting their wealth and status.

The area east of the city near Jian-yang Ward was another densely populated residential district in Northern Wei Luoyang, primarily residential spaces to the general class of commoners. Jian-yang Ward, for instance, housed over 2,000 households, making it the largest residential wards recorded in the Record of the Monasteries of Luoyang. This concentration of commoners underscores the significance of this area within the city's social and residential fabric.

The southern part of the city, known as the Four Barbarians' Lodging House and Four Barbarians' Wards, were residential spaces to foreign envoys and their descendants. According to records, 115 countries sent emissaries to this area, contributing to a vibrant and di-verse environment. This influx of international visitors created a bustling and prosperous scene on the southern side of Luoyang.

Overall, residential spaces in Northern Wei Luoyang were organized by social and ethnic identities. The western side of the outer city was primarily residential spaces to Xianbei royalty, while the eastern side housed Han officials. The southern part of the city was designated for foreign emissaries and their descendants, and commoners were evenly distributed throughout both the eastern and western parts of the city. This spatial organization reflects the distinct social hierarchies of the time.

## Summary of the functional space of the Northern Wei Luoyang City

Northern Wei Luoyang City featured a diverse and complex composition of spaces, with monasteries, government offices, and family residential spaces serving as key functional areas. A comparative analysis of these three space types reveals their distinct yet interconnected roles within the city.

The central axis of Northern Wei Luoyang held a special significance, with a rich spatial sequence running from north to south. Along this axis were the Palace City, administrative offices, monasteries, and residential areas for foreigners. This arrangement highlights the central axis as a vital organizing feature, embodying the city's layered social and functional hierarchy.

The spatial distribution of Northern Wei Luoyang was centered around the Palace City, with government offices located along both sides of the central axis and the east side of the Palace City. Residential areas expanded to the east, west, and south, forming the primary living spaces. Commercial activity was concentrated in key areas, including the east and west sides of the outer city, the central city, and the areas near Four Barbarians' Lodging House and Four Barbarians' Wards. This arrangement highlights the city's structured organization, with distinct zones for administration, residence, and commerce.

In Luoyang, monasteries and residential spaces were generally integrated, with a relatively close spatial distribution. Government offices, however, were concentrated in the inner city, distinctly separated from residential areas. There was also a convergence between official and monastic spaces, as both played roles in representing state power and managing grassroots governance, reflecting their intertwined functions within the city's structure.

The Northern Wei city of Luoyang exhibited clear settlement patterns based on class and ethnicity, creating distinct spatial groups. The inner city was primarily occupied by palaces, government offices, and officials of various nationalities. The western part of the city, considered more prestigious, housed the Xianbei royal family alongside commoners, while the eastern part was mainly residential spaces to Han officials and commoners. The southern and northern sections of the city were less densely populated, with the areas outside the southern city, including Four Barbarians' Lodging House and Four Barbarians' Wards, designated for foreign emissaries and submissive populations. This arrangement highlights the social and ethnic divisions within the city's urban layout.

The overall spatial distribution of Northern Wei Luoyang City follows a circular pattern, with social classes arranged concentrically from the center outward. As one moves outward, the status of the occupants decreases. At the core were the emperor, the imperial family, and high-ranking officials, followed by bureaucrats, then the commoner class, and finally, foreign emissaries and outsiders. This arrangement aligns with the traditional Chinese hierarchical order based on propriety and social rank.

 

## Analysis of the spatial layout of the Northern Wei Luoyang events in the record of the monasteries of Luoyang

The Record of the Monasteries of Luoyang provides locational information alongside the events that occurred around these key nodes. Complex networks allow for the abstraction of real-world elements and their connections into nodes and edges. By utilizing Origin-Destination (OD) data, which records the starting and ending points of characters' movements in these events, we gain valuable insights into urban transportation and the significance of key spatial nodes. This approach supports the investigation of urban flow and the relationships between internal spaces, making it a critical area of urban spatial research.

In this study, the locations of event starting and ending points from the Record of the Monasteries of Luoyang are treated as nodes, while the flow of characters between these locations is represented as edges, forming the basis of a complex network analysis. This method helps to explore the dynamics of urban space in Northern Wei Luoyang.

After thorough reading and filtering, 91 instances of character movement from the Record of the Monasteries of Luoyang were identified. For each event, details such as the starting and ending locations, descriptions of the events, the characters involved, and their identities were recorded to construct a comprehensive Origin-Destination (OD) database of character movement.

These 91 events were categorized into three types: "political activity events", "daily life events", and "auspicious legend events", based on the scenarios that generated the character flow. Complex networks were then constructed for each category and im-ported into software ArcGIS and Gephi for calculation, analysis, and visualization. This approach provides insights into the spatial dynamics of Northern Wei Luoyang.

### Political activities

**Complex network analysis of political activities.** Political activities in Northern Wei Luoyang included events that significantly impacted the social and political landscape, such as the succession of Emperor Zhuang and the coup that killed Er-zhu Rong, who held military power. Additionally, the war-related events, like the Heyin Incident—where Er-zhu Rong executed over 2,000 people—are notable examples. In total, 37 events involving 16 nodes within Luoyang were recorded.

Import the data into Gephi to generate the Complex network map of the political activities (Fig 7). The political activity event network in Luoyang shows that the largest group of nodes, 8 in total, have a degree value of 1. Only 2 nodes have a degree value greater than 10. The Palace City holds the highest degree value, at 14. Overall, the network contains a small number of nodes.

The relevant data, including edges and nodes from the complex network, are matched with the geographic information of the nodes and imported into GIS. Using this data, a map (Fig 8) are generated to visualize the spatial relation-ships and connections within the network.

The map reveals that political activity events were concentrated primarily on the western side of the central axis, near Yong-ning Monastery, Yan-nian Ward(延年里), and Xi-yang Gate(西阳门). Additionally, other events were dispersed around the outer City, creating a single kernel density center. This distribution highlights the centrality of the western region in political activities during the Northern Wei Dynasty in Luoyang.

The degree value distribution map shows that political activities in Luoyang were primarily concentrated in the inner city, particularly on the west side of the central axis and the western part of the city. Only a few nodes extended to the southern outskirts. Key locations within the inner city include the Palace City, Yong-ning Monastery, Yan-nian Ward, and Da-xia Gate. Outside the western city, Shou-qiu Ward stands out as a significant point, while Quan-xue Ward(劝学里) is the most notable location in the southern outskirts. Other areas, such as the northern, eastern, and southern parts of the city—including Four Barbarians' Lodging House and Four Barbarians' Wards—show no distribution of political activity events.

**Fig 7. Complex network map of the Political activities.**

In terms of network node connections, most of the spatial node connections are relatively weak. The strongest connection is between the Palace City and the North side outside Luoyang, significantly surpassing all other nodes, with the heaviest flow be-tween the Palace City and the outer city. Other notable connections include Yan-nian Ward to the Palace City, Shou-qiu Ward to the area outside Luoyang city, the area out-side Luoyang city to Yong-ning Monastery, and the Palace City to Yong-ning Monastery, which show stronger links compared to the rest.

**Discussion of the distribution of political activities.** The Palace City symbolized power in the Northern Wei Dynasty, and access to it represented a direct connection to state authority. As the capital, Luoyang was strongly influenced and controlled by the Palace City, which exerted significant control over the urban space and governance within the city.

The royal family residing in Shou-qiu Ward, on the city's edge, was able to swiftly leave Northern Wei during politically turbulent events. For instance, Some members of the royal family fled to Southern Liang after learning about the Heyin Incident. Quan-xue Ward's importance was further highlighted by the presence of the Imperial College, institutions with significant ritual status. Notable Figures such as Yuan Gong(元恭), who lived in Long-hua Monastery within Shou-qiu Ward before becoming emperor, and Gao Huan(高欢), who took the steles from Shou-qiu Ward when he relocated the capital, underscore the district's prominent role in Luoyang.

The remaining spatial nodes were associated with the imperial family or influential officials. While these Figures played significant roles in Luoyang, they never ascended to the throne, and their power remained dependent on the emperor. Consequently, their spaces within the network were subordinate to the Palace City. True political power rested with the emperor, and neither the Xianbei royalty, Han officials, nor the common people held substantial authority in the Northern Wei Dynasty. As a result, the eastern side of the inner city and surrounding areas were largely absent from the network of political activity events.

Political events in Northern Wei Luoyang primarily occurred between the Palace City and the areas outside of Luoyang. The remaining spatial nodes acted as subsidiary points connected to these two central locations, reflecting their second-ary importance in the political landscape.

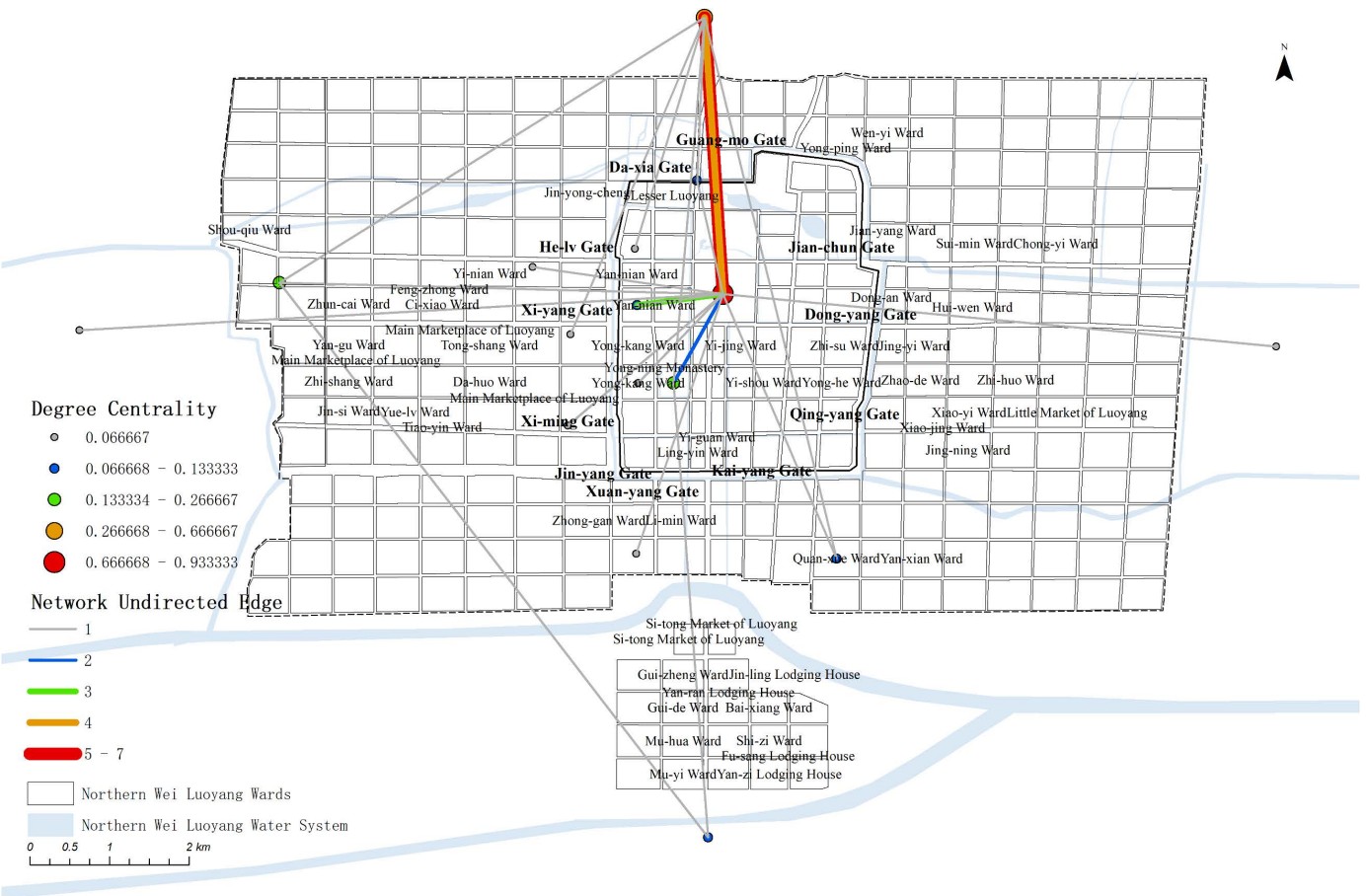

**Fig 8. Map of Political activities complex network.**

### Daily life events

**Complex network analysis of daily life.** Daily life events in Northern Wei Luoyang included various activities across different social classes, such as the emperor's leisure at Hua-lin Park, the daily handling of political affairs, interactions between officials, and annual festivals. These events, while integral to daily life, generally did not have a significant impact on society. A total of 38 events were recorded, involving 34 nodes within the city.

Import the data into Gephi to generate the Complex network map of the daily life events (Fig 9). In the network of daily life events, the majority of nodes (20) have a degree value of 1. The maximum degree value is 14, with only one node having a degree value of 10 or higher. Despite the large number of nodes, 60.6% have a degree value of 1, indicating weaker connections between them. The network's overall connectivity is relatively loose, with many nodes forming smaller, separate networks outside the main structure.

Relevant data, such as edges and nodes from the complex network, were matched with the geographic information of the nodes and imported into GIS. This process was used to generate the map (Fig 10).

Daily life events in Luoyang were concentrated primarily in the inner city, on the west side of the central axis, and near the Chang-he Gate. Additionally, there were a core along the west side of the central axis extending southward and outside the southern city, connected to Four Barbarians' Lodging House and Four Barbarians' Wards. The eastern inner city showed no distribution of daily life events, near Xiao-jing Ward(孝敬里).

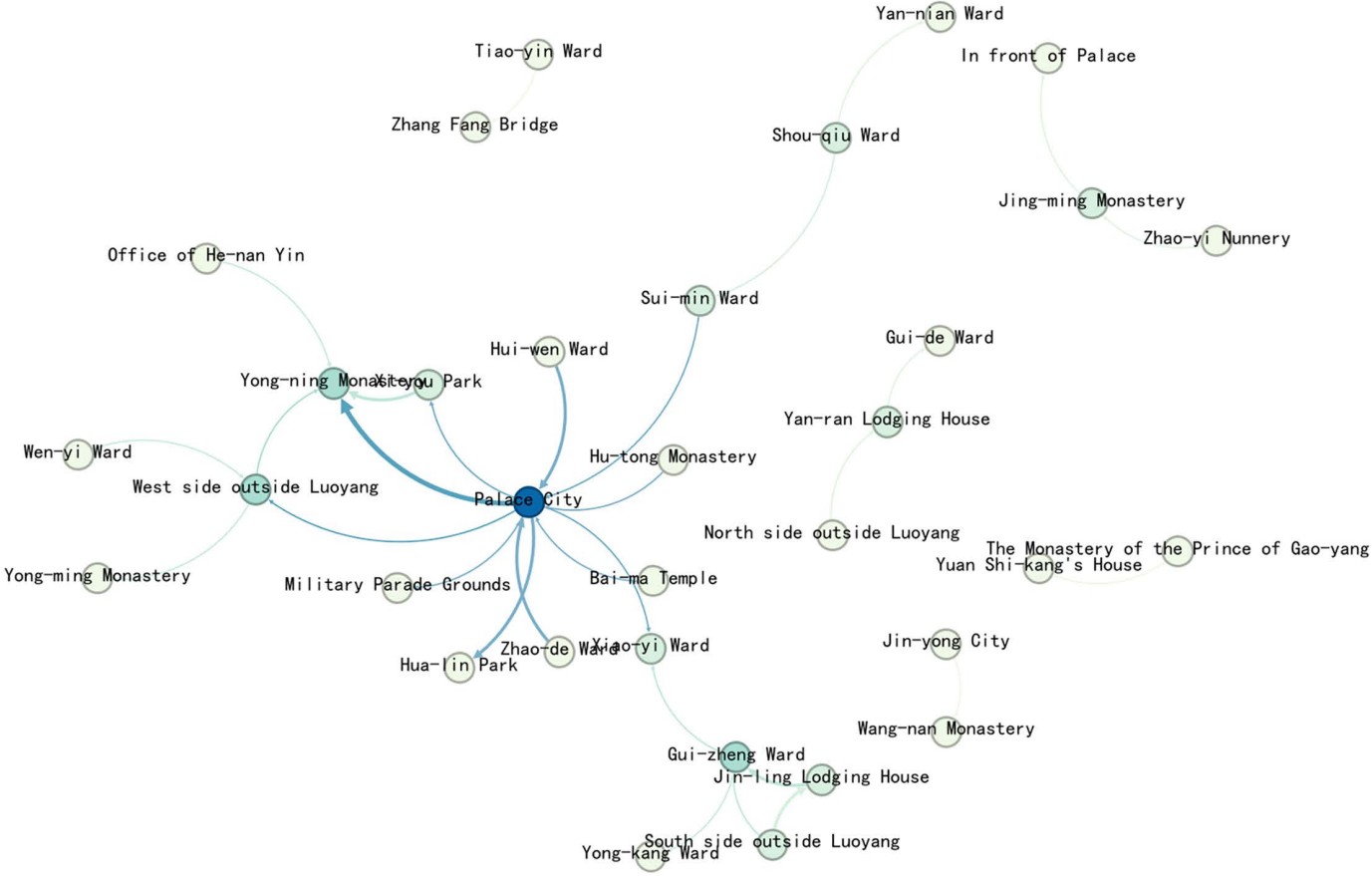

**Fig 9. Complex network map of the Daily life events.**

Regarding the connections between network nodes, the majority of spatial nodes were weakly connected. The strongest connection was between the Palace City and Hua-lin Park, followed by the Palace City and Yong-ning Monastery. Other notable connections included those between Luoyang City Outer South and part of Four Barbarians' Lodging House and Four Barbarians' Wards, and between Luoyang City Outer West and nodes within the city, which showed stronger ties.

**Daily life distribution discussion.** In the main daily life network, the palace was the absolute center of control. Events connected to the palace primarily involved the emperor's daily activities, both in office and at leisure. For instance, Emperor Ming gathered officials to inscribe a dedication for Yong-ning Monastery, visited the temple with Empress Dowager Hu after its completion, rested in Hua-lin Park, and observed military exercises at the parade grounds. These events focus on the emperor's daily life and demonstrate the imperial power's continued dominance over the entire city, with the palace serving as the central hub of the network.

In addition to the main daily life network, there were numerous smaller networks that were not directly connected to the palace. These events typically involved officials or commoners. For example, a General married a dancer from the King Yuan Yong(元雍)'s family after the death of Yuan Yong, and a musician played the reed to send off soldiers at Zhang-fang Bridge(张方桥), boosting their morale. These events, unconnected to the palace, suggest that daily life in Northern Wei Luoyang, as recorded in the Record of the Monasteries of Luoyang, did not always revolve around the palace. It indicates

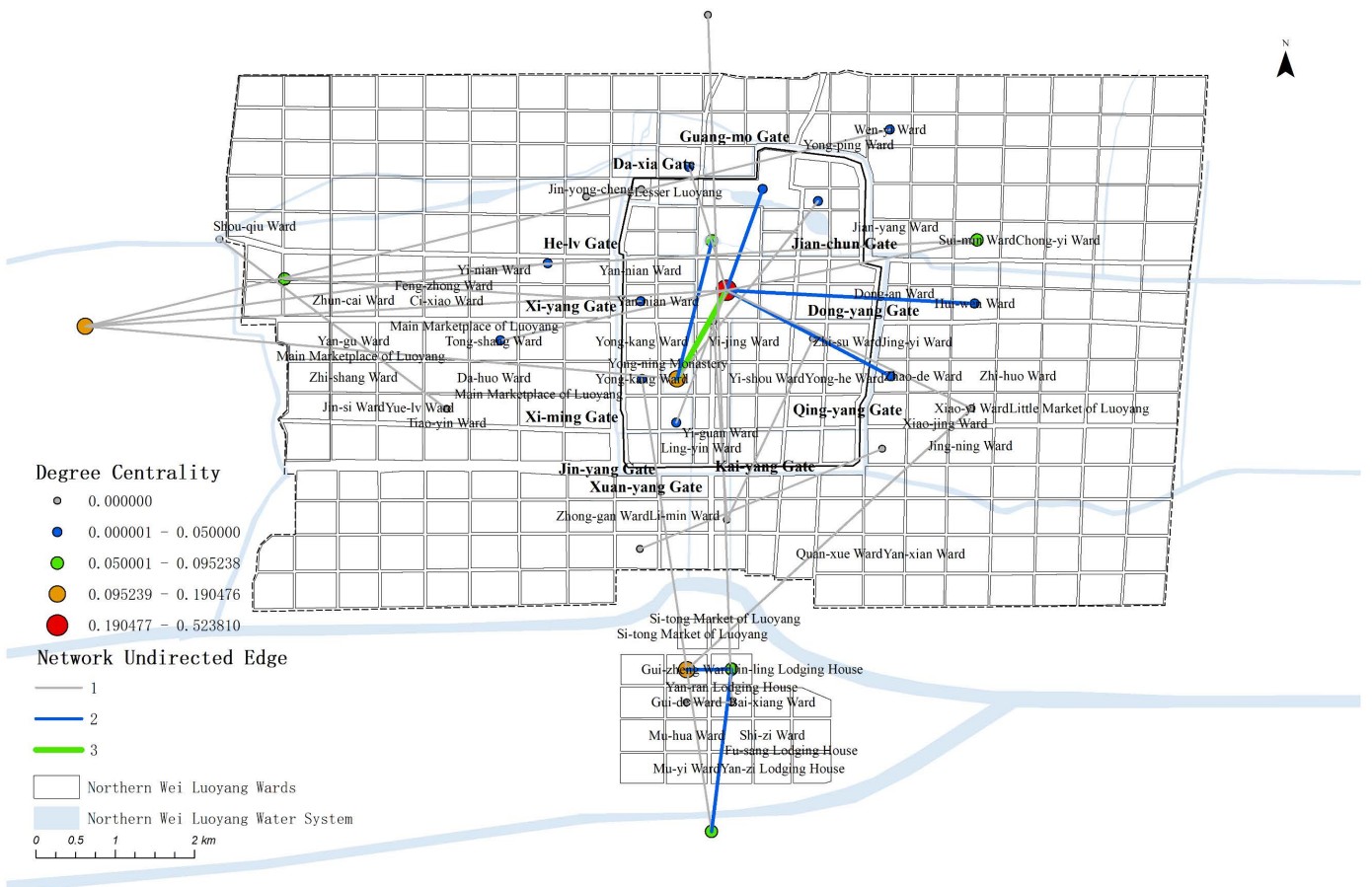

**Fig 10. Map of Daily life events complex network.**

that while the palace had overarching control, common daily interactions and events did not necessarily require palace involvement, allowing for a certain degree of freedom and mobility for the people.

Overall, while the palace dominated daily life in Luoyang, it did not control all locations or events. There was a vibrant exchange of activities among the wards in the city. Southern Liang, in particular, had the highest proportion of visiting states among the Four Directions, and the Northern Wei government maintained a relatively lenient approach in managing outsiders. Additionally, the central axis of Northern Wei Luo-yang played a significant role as the focal point for the city's Buddhist festivals, further emphasizing its importance in the city's social and cultural life.

### Auspicious legend events

**Complex network analysis of auspicious legends.** Auspicious legend events, as recorded in the Record of the Monasteries of Luoyang, include occurrences of unusual or miraculous nature, such as "resurrections from the dead," reflecting the strong religious beliefs of Northern Wei society. A total of 16 events were recorded, involving 15 nodes in Luoyang.

Import the data into Gephi to generate the Complex network map of the daily life events (Fig 11). In the complex network of auspicious legend events, the largest number of nodes—12—have a degree value of 1. The Palace City holds the

highest degree value at 7, while the remaining nodes have degree values of less than 5, showing a clear disparity in the distribution of connections. There are no significant or close connections between the nodes within this network.

The edges and nodes from the complex network, along with related data and node geographic information, were matched and imported into GIS to create the map (Fig 12).

The map shows that the events are primarily concentrated on the east side of the inner city and the eastern part of the city, with less distribution in the west and south. The west side shows an east-west linear distribution along the Chang-he Gate and the Royal Road outside Xi-yang Gate.

In terms of network connections, the Palace City is linked to locations across the entire city, but these connections are generally not very strong. The most significant connection is between Hua-lin Park and Zhun-cai Ward(准财里).

**Discussion of the distribution of Auspicious legends.** Auspicious legend events in Luoyang are richly recorded, though they are often isolated to specific locations and do not form a complex network of characters or actions. Most of these events drew the attention of the emperor or ruling class, with the palace frequently becoming the central point for verification. Rulers often sent envoys to investigate these occurrences, reinforcing the palace's role as the core of auspicious legend events in Luoyang.

Several wards also witnessed auspicious events, most notably Zhun-cai Ward, a district inhabited by the wealthy. Many supernatural events were recorded here, such as a civilian named Cui Han resurrection twelve years after his death, prompting Empress Dowager Hu to send investigators, who confirmed his residence in the area [45] (pp.174–175). Another notable incident involved the ghost of a civilian named Wei Ying visiting his wife after death. These events within Zhun-cai Ward are largely connected to traditional Chinese spirits and monsters [45] (p.205).

Hua-lin Park, although a royal garden with no recorded auspicious events, often alarmed rulers when such events occurred nearby. Ling-tai(灵台), originally a ceremonial structure in Luoyang, was left abandoned during the Northern Wei

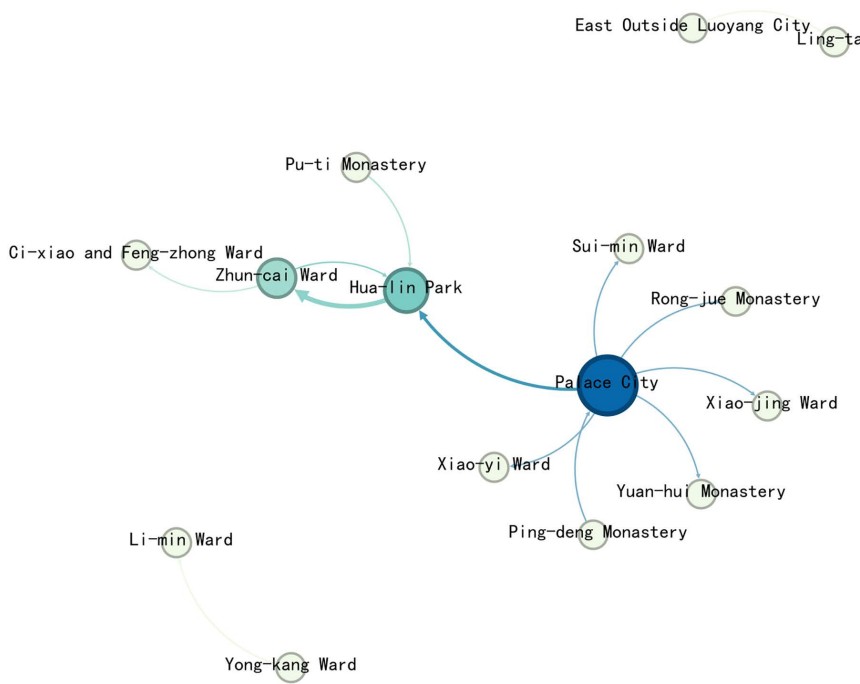

**Fig 11. Complex network map of the Auspicious legend events.**

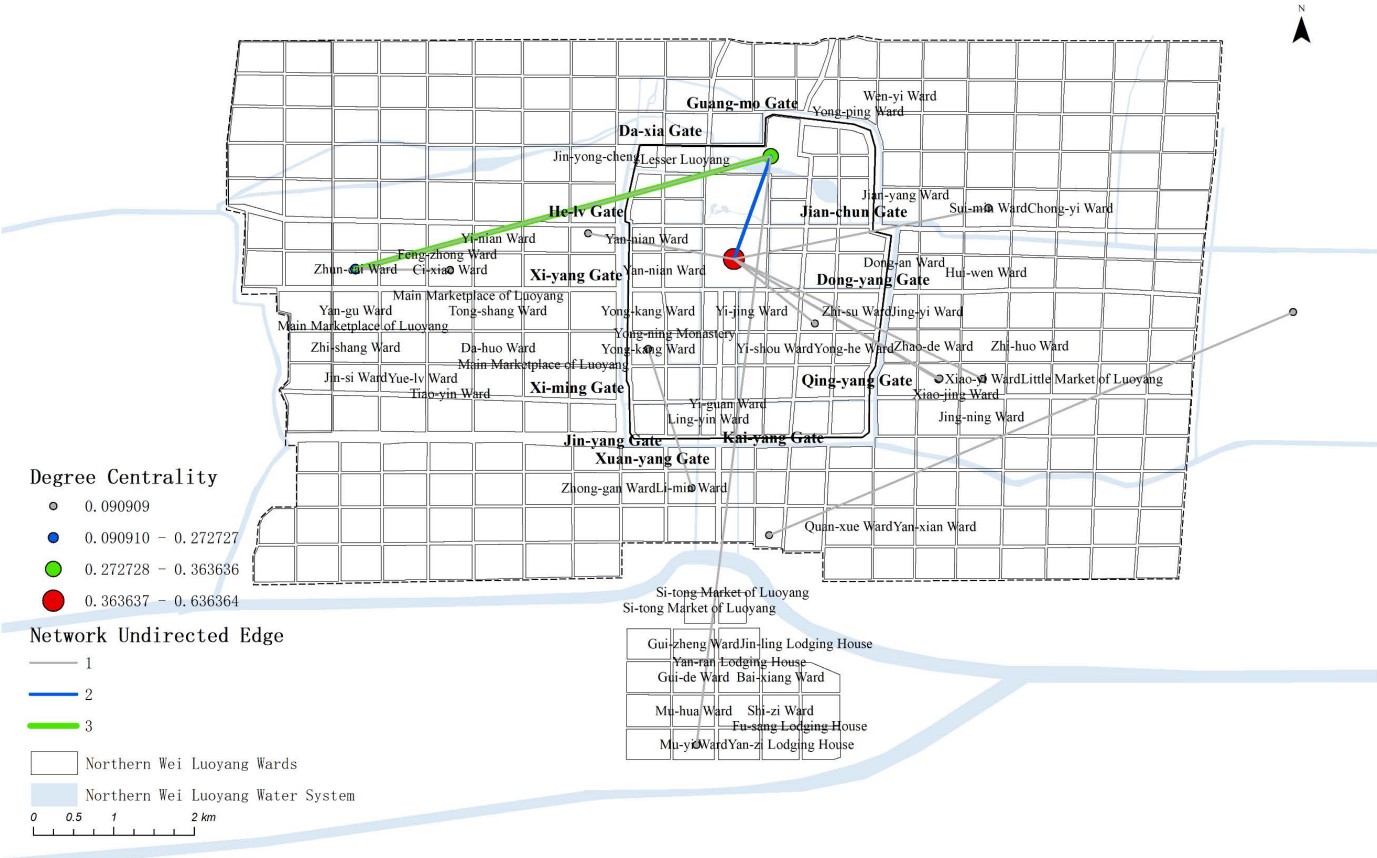

**Fig 12. Map of Auspicious legend events complex network.**

Dynasty. Its proximity to the Luo River, a desolate area, associated it with many of the mysterious and supernatural events recorded in the city.

**Spatial summary of events in the Northern Wei city of Luoyang.** The analysis of the three event types reveals that the event spaces in Northern Wei Luoyang were predominantly centered around royal spaces, such as the Palace City and Hua-lin Park. Most events in Luoyang were closely connected to the Palace City, which exerted control over the entire city, emphasizing its central role in both governance and daily life within the city.

From the perspective of different events, political activities were centered around the Palace City, with most taking place in the western part of the inner city, particularly near the city gates. Daily life events were similarly dominated by the Palace City, with those related to the emperor primarily occurring in the northern section of the inner city's west side. Other daily life events, not directly tied to the emperor, were associated with Han Chinese officials and foreign subordinates, and were distributed across the eastern and southern parts of the city, near Four Barbarians' Lodging House and Four Barbarians' Wards. Auspicious legend events were predominantly concentrated in the eastern part of the city, although Zhun-cai Ward, located in the west, served as a central hub for these occurrences.

The analysis reveals that the Palace City in Northern Wei Luoyang served as the absolute core, playing a crucial role across various events. Additionally, royal spaces such as Yong-ning Monastery and Hua-lin Park, which were exclusive to the emperor, held secondary but significant roles in these events.

In political activities, the most frequent exchanges occurred between the Palace City and the northern outskirts of Luoyang, with a concentration of events near the gates on the western side of the inner city, and very few on the eastern side.

For daily life events, those involving the emperor were centered around the Palace City and distributed in the northern part of the west side of the inner city. Other daily life events, not under imperial oversight, were concentrated on the eastern side of the city and around Four Barbarians' Lodging House and Four Barbarians' Wards in the southern part of the city. Southern Liang was the most significant source of foreigners affiliated with the Northern Wei Dynasty.

In auspicious legend events, most took place in the eastern part of the city. However, Zhun-cai Ward in the west, which had stronger connections to the Palace City, played a more prominent role in these occurrences.

## Discussion

The spatial layout of Northern Wei Luoyang City consisted of different spatial clusters, forming distinct groups. These, along with the central axis, defined the overall structure of the city, with the inner city as the primary hub. The city's layout was centered around the imperial road, with spatial divisions between the east and west, while the four corners of the city were more sparsely populated.

In Northern Wei Luoyang, the eastern and western parts of the city formed distinct groups. The northern side, due to its terrain, did not develop a concentrated group, while the southern group extended along the central axis down to Four Barbarians' Lodging House and Four Barbarians' Wards. The inner city held greater importance compared to the outer city, with the western part of the outer city being more prominent than the eastern part. This spatial organization highlights the city's hierarchical and centralized structure.

### Layout characteristics of Northern Wei Luoyang

**Layout characteristics of the single axis reinforcement in Northern Wei Luoyang.**  During the Northern Wei Dynasty, the central axis of Luoyang was further emphasized, becoming one of the city's core spaces. It began at the northern end of the Palace City, extending southward through the Palace of the Great Ultimate and the Chang-he Gate, with key state institutions lining both sides within the inner city. On the southern side, prominent monasteries were distributed along the axis, eventually reaching Four Barbarians' Lodging House and Four Barbarians' Wards outside the southern city limits. Although the central axis was located to the west of the inner city, it served as the central dividing line in the outer city, effectively splitting Luoyang into eastern and western halves(Fig 13). Thereafter, the triple-walled structure and central axis became defining features of ancient Chinese cities. As imperial power grew stronger, these characteristics were further reinforced: the central axis gained prominence, with increasingly higher-ranked buildings flanking it, and the axis itself was extended. The inner city was transformed into the emperor's exclusive imperial city. By the time of Ming Beijing(16th century CE), an additional layer of city walls was constructed on the southern side.

Strengthening of axis sequence nodes: In the Eastern Han period, Luoyang's urban layout featured a pattern of north and south palaces side by side, with multiple axes that did not occupy a central position in the city. However, during the Wei and Jin periods, the city transitioned to a single axis and palace layout, establishing the central axis as a defining feature of Luoyang's spatial organization. Under the Northern Wei Dynasty, the significance of the central axis was further reinforced, with key nodes and spaces arranged along its length.

In Northern Wei Dynasty Luoyang, the palace city held a central spatial position within the urban layout and served as the focal point for various events within the city. This dual centrality solidified its role as the most significant node in the city's structure, further reinforcing the prominence of the central axis that traversed the palace city. From the perspective of spatial sequences and events along the central axis, this area became crucial for promoting Northern Wei culture, showcasing national power, and serving as the primary venue for Buddhist activities. The central axis thus emerged as a focal point for both political and religious life in the city.

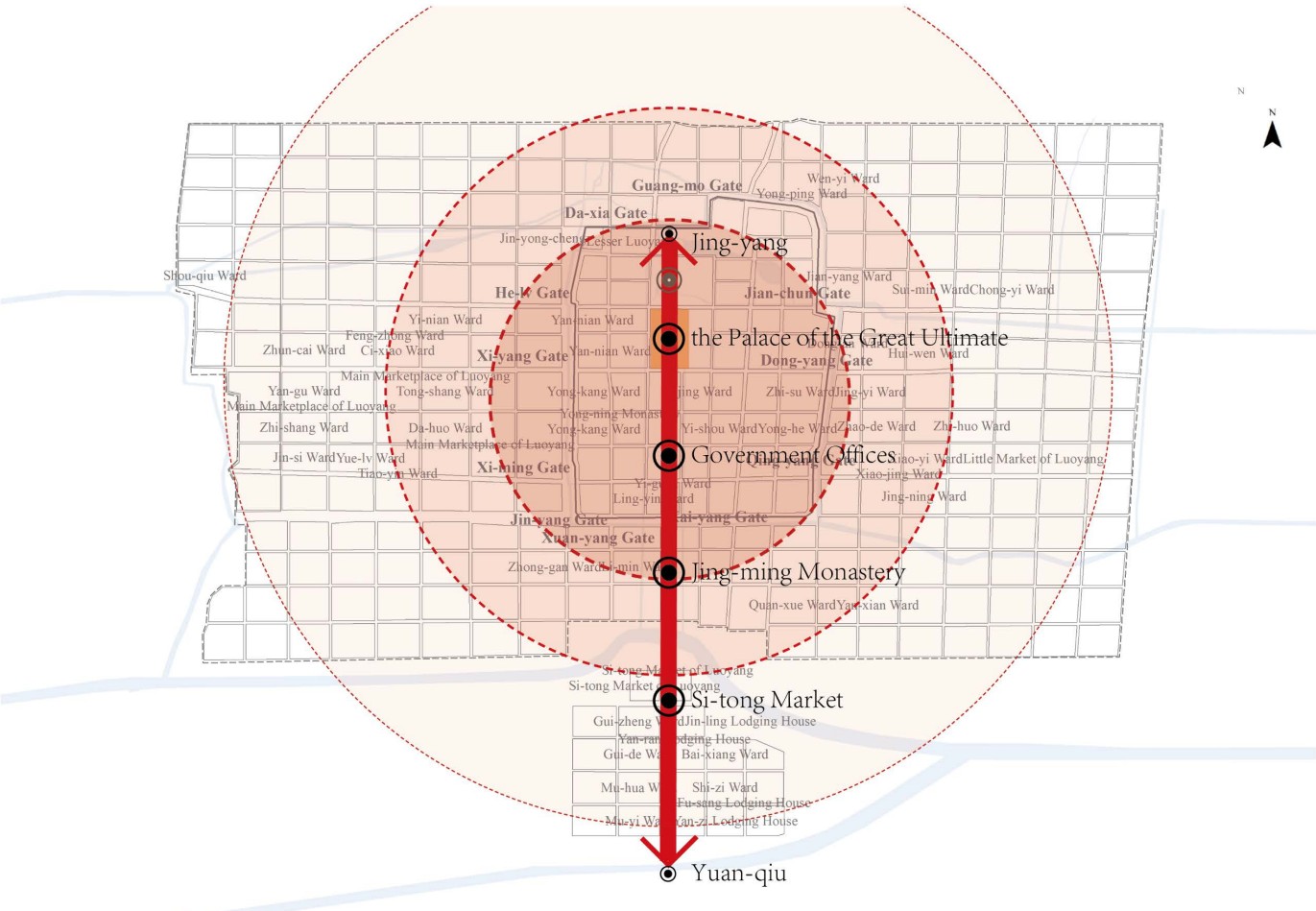

**Fig 13. Layout characteristics of the single axis reinforcement.** During the Northern Wei period, the central axis of Luoyang was further strengthened, and its role in the city's spatial distribution became more pronounced.

**Establishment of the outer city:** The Northern Wei Dynasty established the Outer City around Luoyang's inner city, using the central axis as its symmetrical foundation. This involved relocating the round mound and extending the central axis southward to its vicinity. At the southern end of the axis, the Four Barbarians' Lodging House and Four Barbarians' Wards were constructed, further reinforcing the central axis as the primary organizing feature of the city. This development enhanced the prominence of the central axis, establishing it as the dominant spatial and symbolic element in Luoyang's urban layout.

**Circularized layout along the axis:** The spatial layout of Northern Wei Luoyang reflects a stratified organization along the central axis. The spaces on either side of the axis are symmetrically arranged, with class distinctions gradually diminishing from the center outward. The highest density of space is near the axis, where state power institutions are located on both sides within the inner city. Beyond these are areas occupied by the Xianbei royal family, Han officials, and other elite classes. Further outward, the space is allocated to commoners, demonstrating a clear hierarchy in spatial distribution based on class, with density decreasing as one moves away from the central axis.

**The formation of the single axis reinforcement in Northern Wei Luoyang is closely tied to the continuation of the original urban spatial framework and the influence of urban design principles derived from Northern Wei Pingcheng.**

**Continuation of the original urban space:** The construction of Northern Wei Luoyang was deeply rooted in the urban planning practices of previous dynasties. The spatial layout of the city was heavily influenced by the Eastern Han Dynasty, as well as the Wei and Jin Dynasties. For instance, the axial arrangement, such as the multiple axis in Eastern Han Luoyang, set a precedent that was further reinforced during the Wei and Jin periods under the guidance of *Kao gong ji* (考工记), an influential ancient Chinese construction manual.

Northern Wei planners inherited and expanded upon these precedents, integrating traditional Central Plains rituals and etiquette into their designs. This led to the symmetrical construction of the Outer Kuo City along the central axis, emphasizing the city's hierarchical structure. Emperor Xuanwu further extended the axis by relocating the Round Mound to the north of the Yi River and constructing a pontoon bridge across the Luo River [47]. This extension connected the central axis to the Round Mound and expanded Luoyang's spatial scope. Additionally, Emperor Xuanwu established the Four Barbarians' Lodging House and Four Barbarians' Wards between the Yi and Luo Rivers to accommodate visitors and delegations from outside regions, further underscoring the central axis's importance while reinforcing Luoyang's role as a cultural and administrative hub.

This deliberate reinforcement of the central axis demonstrates the Northern Wei Dynasty's commitment to continuity in urban design while adapting and extending earlier traditions to create a more expansive and sophisticated cityscape.

**Influence of the urban form of Pingcheng in the Northern Wei Dynasty:** Before relocating the capital to Luoyang, the Northern Wei Dynasty established its capital in Pingcheng (modern-day Datong, Shanxi Province, China). The design of Pingcheng was influenced by the ritual systems of the Central Plains, adopting and refining the single-palace system seen in cities such as Yecheng and Chang'an [60]. This design emphasized the central axis as a defining feature of the urban form.

The experience gained from constructing and organizing Pingcheng played a significant role in shaping the urban planning of Luoyang after the capital's relocation. The spatial layout of Pingcheng, with its focus on axial symmetry and hierarchical organization, influenced the internal structure and overall spatial arrangement of Luoyang. These principles were carried forward to reinforce Luoyang's central axis, ensuring continuity in urban design while integrating the ritualistic and practical considerations central to Northern Wei governance and cultural traditions.

**Layout characteristics of multiple cluster divisions in Northern Wei Luoyang.** In Northern Wei Luoyang, various spatial groupings emerged due to the distribution of different functions, events, and social classes. These spatial clusters clearly reflect class distinctions, with the status of the individuals occupying them gradually decreasing from the center of the city outward. The inner city, closer to the central axis, was occupied by higher classes, while the outer areas were reserved for lower-status groups, highlighting the hierarchical nature of the city's spatial organization(Fig 14). Although the disintegration of the ward system eventually led to the decline of strictly defined spatial planning in ancient Chinese cities, the hierarchical spatial structure, characterized by a gradual decline in status from the center outward, became more pronounced. Meanwhile, commercial activities retained their characteristic organization, with specific industries concentrated in designated areas and remaining relatively separated from one another.

**Differentiation and aggregation based on identity in the spatial clusters of Northern Wei Luoyang:** The spatial layout of Northern Wei Luoyang reflects distinct clusters differentiated by function, class, and other factors. The organization of the city follows the principle of "class-based spatial distribution," with both the inner and outer city exhibiting unique characteristics of social stratification.

In the inner city, clusters were primarily differentiated by class and official position rather than ethnicity. The palace served as the central focal point, with the emperor at its core. Proximity to the palace and the central axis was a key marker of status—higher-ranking officials and those of higher social classes lived closer to these central areas.

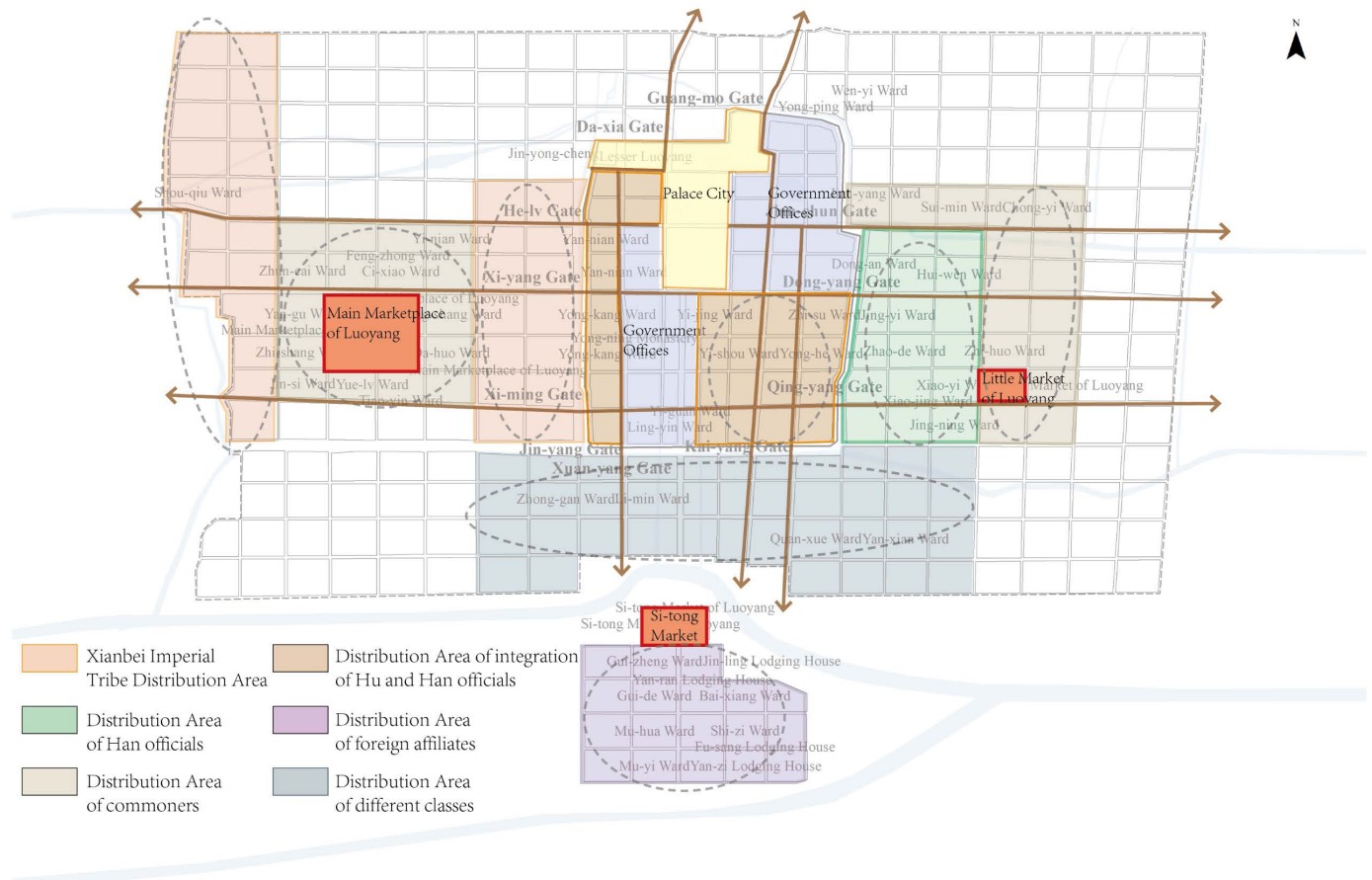

**Fig 14. Layout Characteristics of Multiple Cluster Divisions.** The Multiple Cluster Divisionsin Northern Wei Luoyang was shaped by several factors, including the dominance of the Hua-yi distinction (Sino–barbarian dichotomy), the ethnic distinctions within the population, and the development of commercial spaces.

In the outer city, groups were organized based on both ethnicity and class. The western part, closest to the inner city, was predominantly occupied by the Xianbei royal family, with artisans and craftsmen residing in the outer areas, near the Main Market-place of Luoyang. At the very edge of the outer city, the extended royal family of the Northern Wei Dynasty resided. In contrast, the eastern part of the city was primarily residential spaces to Han officials and commoners. The southern part, near Four Barbarians' Lodging House and Four Barbarians' Wards, was mainly inhabited by foreign groups who had returned to the city.

This stratification was particularly noticeable in Main Marketplace of Luoyang's on the western side of the city, where living spaces were organized by occupation. For instance, funeral practitioners resided on the north side of Main Market-place of Luoyang, while brewery practitioners lived on the west side. This arrangement further highlights the city's systematic organization based on class and profession.

**Dominance of the hua-yi distinction:** The Northern Wei Dynasty, established by the Xianbei—a non-Han ethnic group—adopted the traditional Chinese hierarchical order to integrate into the Central Plains and promote the Sinicization of the Xianbei people. This system emphasized the cultural superiority of Han Chinese while maintaining distinctions between

groups. The spatial organization of Luoyang reflected these principles, with the emperor situated at the center, surrounded by officials, commoners, and foreign subservient groups in increasingly peripheral zones.

This hierarchical spatial arrangement extended to the commercial areas of the city, where practitioners were segregated by occupation, ensuring alignment with the ideology of "grouping by class. " Ethnic and social groups occupied distinct urban clusters, reinforcing both functional and symbolic separations within the city. This design not only facilitated governance but also mirrored the sociopolitical structure of the Northern Wei Dynasty, balancing traditional Chinese practices with the integration of the Xianbei ruling elite.

These factors contributed to a complex and layered urban form, characterized by distinct spatial zones for different social and ethnic groups, and established a precedent for later urban designs in Chinese history.

**Distinction of different ethnic groups:** The residents of Northern Wei Luoyang's inner city came from diverse ethnic backgrounds, primarily consisting of the Xianbei and other non-Han groups, as well as Han Chinese. Within the inner city, under the emperor's authority, the principle of Hua-yi distinction was implemented, promoting the integration of Hu (non-Han) and Han populations. However, in the outer city, the ethnic groups were more segregated, maintaining spatial and social distinctions. These ethnic divisions significantly influenced the differentiation of social groupings and the organization of urban space.

**Development of urban commercial space:** In contrast to earlier dynasties, Northern Wei Luoyang featured a more systematic establishment of commercial districts across the outer city. Major markets were strategically located to cater to different needs: the Main Marketplace of Luoyang was established in the west, the Little Market of Luoyang in the east, and the Si-tong Market in the south, near the Four Barbarians' Lodging House and Four Barbarians' Wards. These markets were not only geographically distributed but also functionally organized, with craftsmen and commercial workers spatially separated from other classes. This deliberate grouping reinforced the city's social and economic stratification, creating distinct zones for commerce and trade while contributing to the multi-cluster layout of the city.

**Layout characteristics of the prestige of the West in Northern Wei Luoyang.** In Northern Wei Luoyang, the central axis served as the dividing line, separating the inner and outer city into distinct eastern and western sections. This division emphasized the greater importance of the western side of the city in both physical and social terms (Fig 15). However, not all Chinese cities throughout history with a symmetrical central axis exhibited this characteristic. In mature examples of Chinese urban spatial structures, such as Beijing during the Ming and Qing dynasties, the differences between the eastern and western sides of the central axis were minimal. The westward-prestige layout may have been a unique phenomenon of the Northern Wei period.

Relative Importance of the Western Side of the Inner City: The Palace City on the northern side was the most critical core space within the inner city. However, the inner city was further divided by the central axis, and the western side was regarded as more important than the eastern side. The western side of the central axis was primarily occupied by high-ranking officials and royal family members who held significant state power. For instance, Yong-ning Monastery, built by Empress Dowager Hu's relatives, was located on this side. In contrast, the eastern side was residential spaces to relatively lower-ranking officials. Spaces like Yong-he Ward were primarily occupied by officials of the third or fourth rank, and the nearby Zhao-yi Nunnery was funded by ordinary eunuchs, reflecting the more modest status of the east.

Western Nobility in the Outer City: The preference for the western side extended to the outer City as well. The western part of the city was where most of the imperial family resided and where most political events occurred. In contrast, the eastern part was populated by Han Chinese officials, but it was largely devoid of political events. The political happenings in Luoyang, par-ticularly those related to shifts in power, reveal that the Xianbei royals were more closely tied to the power structures of Northern Wei than their Han counterparts.

In terms of daily life events, the eastern part of the city was relatively independent and had fewer connections with the palace. Furthermore, the Main Marketplace of Luoyang was located in the western part of the city, which was larger and more prosperous than the smaller marketplace in the east, reinforcing the prominence of the western side of the city.

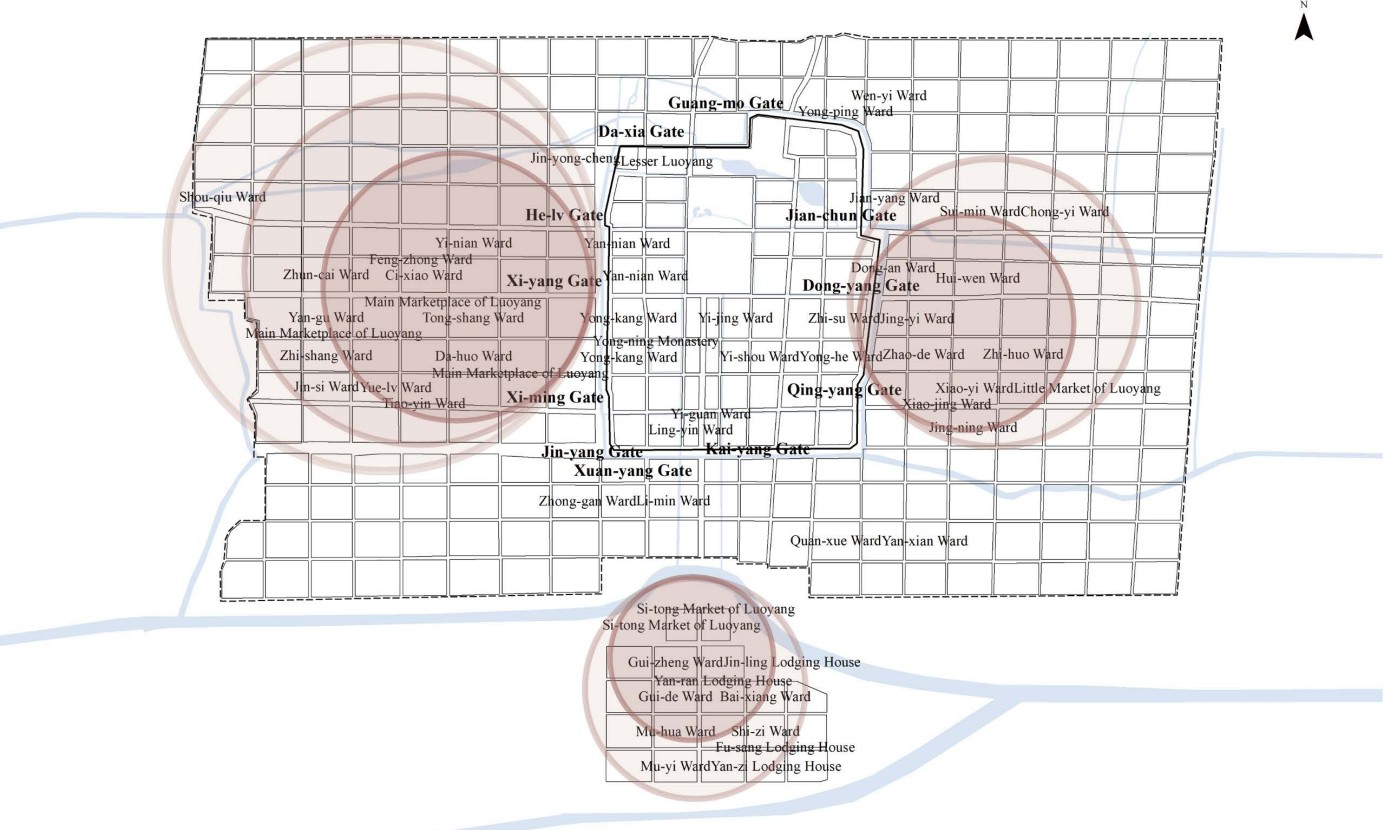

**Fig 15. Layout Characteristics of the Prestige of the West.** The western area of Northern Wei Luoyang was considered prestigious for several reasons, most notably the continuation of traditional Xianbei customs and the influence of the city's geographic location.

**Inheritance of Traditional Xianbei Customs:** In Xianbei culture, the west held a position of great significance, a tradition rooted in their nomadic heritage and sacrificial rituals. According to the *Book of the Later Han – Treatise on the* Wuhuan*, Xianbei (后汉书·乌桓鲜卑列传), Their architectural practices typically oriented buildings eastward, symbolizing reverence for the sun and the west as the most sacred direction, the Xianbei people viewed the west as the most esteemed direction [61].

This cultural belief carried over to Pingcheng, the former capital of the Northern Wei Dynasty, where the emperor conducted sacrifices to the heavens in the western suburbs, and various social classes also directed their worship westward. However, following the relocation of the capital to Luoyang in 494 CE, Emperor Xiaowen abolished the western suburban sacrifices to the heavens, replacing them with Han Chinese ritual practices [47,50]. Despite this shift, the reverence for the "Hu God" (a Xianbei deity) persisted, suggesting that the west retained its symbolic importance [62].

This cultural tradition may explain why the Northern Wei imperial family and elites predominantly occupied the western section of Luoyang's outer city. Even as they adopted Han rituals, the enduring symbolic status of the west in Xianbei culture influenced urban spatial organization, reinforcing the prominence of the western area in Luoyang.

This cultural and historical context highlights how the integration of Xianbei traditions and Central Plains rituals shaped the city's layout and social hierarchy.

**The influence of Luoyang's geographic location:** The geographic location of Northern Wei Luoyang played a crucial role in shaping the city's spatial dynamics and social organization, particularly the prominence of its western side. As described in the *Essence of Historical Geography or Essentials of Geography for Reading History – Henan* (读史方舆纪要·河南志), Luoyang was considered "more beautiful than any other city in the world" [63], with Mount Mang to the north and Yique to the south. Its location at a key transportation junction between the Guanzhong Plain and the Central Plains made it an important spatial node for controlling east-west exchanges, enhancing its strategic significance [64].

Prior to Luoyang becoming the capital, the Northern Wei Dynasty's capital was located in Pingcheng. The geographic constraints of the region necessitated that large numbers of troops and personnel traveling from Pingcheng to Luoyang pass through the relatively flat terrain of the Jinzhong and Yuncheng Basins, as other routes were dominated by mountainous landscapes unsuitable for large-scale movement. This route brought travelers and resources into Luoyang from the west, naturally establishing the western side of the city as the dynasty's logistical and administrative rear. Conversely, the eastern side of Luoyang faced the Central Plains and the Southern Dynasties, a region that posed greater risks, particularly during potential military conflicts with the Southern Liang.

As a result, from the very beginning of Luoyang's establishment as the capital, the western side was primarily occupied by the imperial family. Historical events underscore the strategic importance of this area. When the Northern Wei Dynasty eventually split, Emperor Yuan Xiu fled westward to Xi'an and founded the Western Wei Dynasty, while Gao Huan, a key minister, moved eastward to Yecheng to establish the Eastern Wei Dynasty. These developments highlight the enduring significance of the western side of Luoyang, while the eastern side became the domain of officials and Han scholars.Ironically, the greatest challenge to the Northern Wei Dynasty's royal family and officials, the Heyin Rebellion, originated from Erzhu Rong, a figure from Xiuyong (modern-day Xinzhou, Shanxi Province).

## Limitations of data sources

Despite the insights gained in this study, the data used have certain limitations that could affect the conclusions.

First, the spatial points analyzed in this paper are limited to those with detailed records in the *Record of the Monasteries of Luoyang*. Due to constraints in the text and the author's interpretation, the dataset may not fully represent all spatial points in Northern Wei Luoyang. This incomplete dataset could result in discrepancies between the actual spatial layout and the patterns inferred from the analysis.

Second, the accuracy of the reconstructed maps and points presents another challenge. While some locations are corroborated by archaeological evidence, the spatial information recorded in ancient texts, such as the *Record of the Monasteries of Luoyang*, is accurate only to the level of wards. Comparisons between textual descriptions and archaeological findings for key nodes, such as Yong-ning Monastery and the Palace Gate, suggest a high degree of alignment, indicating that the text is reasonably accurate. However, the precise positioning of certain locations requires further verification through future archaeological discoveries.

Additionally, the modern version of the *Record of the Monasteries of Luoyang* originates from a 16th-century Ming Dynasty edition. Over more than a thousand years of transmission, the text may have undergone alterations, potentially deviating from its original form. However, historical analyses and verifications of the text by scholars lend credibility to its accuracy, allowing it to serve as a reliable data source for this study.

Lastly, the information in the *Record of the Monasteries of Luoyang* reflects the locations of significant sites associated with powerholders and merchants, providing limited insights into the spaces occupied by commoners. While this focus may not fully capture the broader social dynamics of Luoyang, it does offer a clear depiction of the city's axes, clusters, and east-west spatial differences, which are crucial for understanding its overall layout during the Northern Wei Dynasty.

In conclusion, while the data have limitations, they provide a valuable framework for analyzing the spatial characteristics of Northern Wei Luoyang and serve as a foundation for future research and archaeological validation.

 

## Conclusion

This study extracts, reconstructs, and visualizes historical information using digital technology, providing a novel approach for studying underground historical cities. By focusing on the *Record of the Monasteries of Luoyang*, combining archaeological data, and applying tools such as historical GIS and complex network analysis, the study identifies and organizes spatial nodes and related events recorded in the text. It further summarizes the layout characteristics of Northern Wei Luoyang based on three functional spaces (monasteries, government offices, and residential spaces) and three event spaces (political activities, daily life events, and auspicious legend events). The findings reveal the influence of factors such as geography, urban development, Central Plains rituals, traditional customs, and ethnic diversity on the formation of Luoyang's spatial layout. This study offers innovative and valuable insights for understanding the Northern Wei Dynasty city of Luoyang's structure and its implications for other underground historical cities.

However, certain aspects of the *Record of the Monasteries of Luoyang* remain underutilized. Data such as the heights of temple towers, details of gardens and their flora and fauna, and records of human interactions in less-documented locations offer potential for further research. The study also does not incorporate temporal dynamics due to the limited chronological scope of the text. Future research could combine multiple historical texts and archaeological evidence to explore spatial and temporal patterns more comprehensively.

In terms of methodology, this study partially employed automated text extraction. In the future, advanced techniques such as corpus linguistics, natural language processing, and AI could be applied to enhance the analysis of historical texts.

Finally, this research provides valuable data and visualizations that can inform heritage conservation and site restoration efforts for Northern Wei Luoyang. These findings can support further research, planning, protection, and public presentation of the site, ensuring its historical significance is preserved and better understood for future generations.

## Supporting information

**S1 File. Pictures of Northern Wei Luoyang Site.**
(PDF)

**S2 File. Tables of Filtered Data.**
(ZIP)

**S3 File. Functional Space Data.**
(ZIP)

**S4 File. Functional Space Input Parameter.**
(XLSX)

**S5 File. Functional Space Spatial Analysis Result.**
(ZIP)

**S6 File. Complex network Files.**
(ZIP)

**S7 File. Complex Network Tables with Position Information.**
(ZIP)

## Acknowledgments

The authors would like to express my sincere gratitude to Prof. Guoxiang Qian, Dr. Wei Zuo, Dr. Yale Ye, Geng Pei and Zekai Li for their valuable support in software development and literature search, which greatly contributed to the success of this work.

## Author contributions

**Conceptualization:** Runfeng Sun, Wenjia Liu.

**Data curation:** Runfeng Sun.

**Formal analysis:** Runfeng Sun.

**Funding acquisition:** Wenjia Liu.

**Investigation:** Runfeng Sun, Wenjia Liu.

**Methodology:** Wenjia Liu.

**Project administration:** Wenjia Liu.

**Resources:** Wenjia Liu.

**Software:** Runfeng Sun.

**Supervision:** Wenjia Liu.

**Validation:** Wenjia Liu.

**Visualization:** Runfeng Sun.

**Writing – original draft:** Runfeng Sun.

**Writing – review & editing:** Runfeng Sun, Wenjia Liu.

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
