## [Decision Letter · Decision Letter 0]

29 Nov 2024

PONE-D-24-46751Spatial Layout Characteristics of Northern Wei Luoyang: A Spatial Humanities Analysis of the “Record of the Monasteries of Luoyang”PLOS ONE

Dear Dr. Liu,

Thank you for submitting your manuscript to PLOS ONE. After careful consideration, we feel that it has merit but does not fully meet PLOS ONE’s publication criteria as it currently stands. Therefore, we invite you to submit a revised version of the manuscript that addresses the points raised during the review process.

We look forward to receiving your revised manuscript.

Kind regards,

Yuan Zhang, PhD

Academic Editor

PLOS ONE

Journal requirements: When submitting your revision, we need you to address these additional requirements. 1. Please ensure that your manuscript meets PLOS ONE's style requirements, including those for file naming. The PLOS ONE style templates can be found at https://journals.plos.org/plosone/s/file?id=wjVg/PLOSOne_formatting_sample_main_body.pdf and https://journals.plos.org/plosone/s/file?id=ba62/PLOSOne_formatting_sample_title_authors_affiliations.pdf 2. Please amend either the title on the online submission form (via Edit Submission) or the title in the manuscript so that they are identical. 3. Please match your authorship list in your manuscript file and in the system. 4. Please include a caption for figure 3, Fig 4, Fig 5, Fig 6, Fig 7, Fig 8, Fig 9, Fig 10 and Fig 11. 5. When completing the data availability statement of the submission form, you indicated that you will make your data available on acceptance. We strongly recommend all authors decide on a data sharing plan before acceptance, as the process can be lengthy and hold up publication timelines. Please note that, though access restrictions are acceptable now, your entire data will need to be made freely accessible if your manuscript is accepted for publication. This policy applies to all data except where public deposition would breach compliance with the protocol approved by your research ethics board. If you are unable to adhere to our open data policy, please kindly revise your statement to explain your reasoning and we will seek the editor's input on an exemption. Please be assured that, once you have provided your new statement, the assessment of your exemption will not hold up the peer review process.

Reviewers' comments:

Reviewer's Responses to Questions

**Comments to the Author**

1. Is the manuscript technically sound, and do the data support the conclusions?

Reviewer #1: Yes

Reviewer #2: Partly

2. Has the statistical analysis been performed appropriately and rigorously? 

Reviewer #1: N/A

Reviewer #2: I Don't Know

3. Have the authors made all data underlying the findings in their manuscript fully available?

Reviewer #1: Yes

Reviewer #2: No

4. Is the manuscript presented in an intelligible fashion and written in standard English?

Reviewer #1: No

Reviewer #2: Yes

5. Review Comments to the Author

Reviewer #1: The manuscript presents a digital approach to analyzing historical urban spaces using spatial humanities tools, specifically GIS and complex network analysis. The focus on Northern Wei Luoyang provides a valuable case study that contributes to the understanding of ancient urban planning and social stratification.

A few suggestions are as the following:

(1) Include a critical discussion of the results, considering potential alternative interpretations or explanations for the observed patterns to provide a more nuanced understanding.

2�Discuss potential limitations or biases in the data sources used and how these might affect the study's conclusions.

3�the paper has explained more “what”, while less interpretation of “why” and “how”, namely the reason behind the patterns.

(4) the conclusion part is rather weak in the present form.

Reviewer #2: - What are the main claims of the paper and how significant are they for the discipline?

This is a comprehensive study of urban history applied to a city in China. It mixes quantitative and qualitative methods effectively and the discussion of results is convincing. I have no expertise to evaluate the subject matter as I am neither a a urban historian nor an expert on China. From a the perspective of a scholar engaged in spatial humanities projects and as a geographic information scientist, I found the study well conducted if not especially innovative. The use of the GIS and the spatial analytical methods applied are appropriate to address the research questions, but fairly common, as is the complex network analysis. This is not a criticism per se, but the claim in the abstract (and implied elsewhere) that this is a "new digital framework" is unwarranted. While the application area may be novel, something I am in no position to judge, the methods and the analytical framework certainly are not. As a final note, the maps and figures in the manuscript are well done and meaningful.

- Are the claims properly placed in the context of the previous literature? Have the authors treated the literature fairly?

I cannot comment on the topical literature (i.e., urban history of China) but the spatial humanities and also the network analysis literature are not up to date. For example, the authors may have benefited from applying principles and methods from social spatial network analysis (see in particular the work of Clio Andris and others on the subject). A deeper dive in the literature would be welcome but is not necessary, as the analytical tools used by the authors are quite standard and appear sufficient to justify their conclusions. As a suggestion, perhaps for future work, in addition to spatial social networks, the use of corpus linguistics, natural language processing, and even AI could have been used here.

- Do the data and analyses fully support the claims? If not, what other evidence is required?

On the surface, yes. But the data have not been provided so I am unable to judge.

- Are original data deposited in appropriate repositories and accession/version numbers provided for genes, proteins, mutants, diseases, etc.?

No, although the authors state that data will be provided "upon acceptance of the manuscript."

- Are details of the methodology sufficient to allow the experiments to be reproduced?

No, at least as concerns spatial analysis. No details are provided on the parameters entered for kernel density analysis and for the calculation of mean centers and standard deviation ellipses. They should.

- Is any software created by the authors freely available?

ArcGIS is not, but I believe the other programs mentioned are freely available.

- Is the manuscript well organized and written clearly enough to be accessible to non-specialists?

The manuscript is well organized and very clearly written. There are several typos and other minor issues, including in no particular order: a) spelling mistakes and typos include, but are not limited to, "fortyyear" (line 80), wrongly spaced quotation marks (110, 11, and elsewhere), "highdefinition" (120) "ex-ample" (276), "con-centration" (299), etc. Also, no need to write "Mr." in line 102 and "was" in line 301 is grammatically incorrect in line 301; b) "Geographic Information Systems (GIS)" should be spelled out only the first time it appears. Everywhere else just write "GIS"; c) GIS is called a "method" in the abstract but a technology elsewhere (e.g., line 32). GIS is a tool or a technology, not a method (Kernel density is a spatial analytical method); d) the content of the paragraph between line 142-145 is repeated immediately below in lines 146-150 with slightly different wording, keep only one; e) I would replace the word "alignment" with "georeferentiation" in line 151; f) the term "spatialized map" (369 and elsewhere) is redundant, just write "map"; g) the term "spatial location" (340) is also redundant, just use "locational"; h) the term "like" in line 355 is not needed.

6. PLOS authors have the option to publish the peer review history of their article (what does this mean? ). If published, this will include your full peer review and any attached files.

**Do you want your identity to be public for this peer review?** For information about this choice, including consent withdrawal, please see our Privacy Policy .

Reviewer #1: No

Reviewer #2: No

---

## [Author Response · Author response to Decision Letter 1]

19 Jan 2025

Response to Academic Editor

Dear Dr. Zhang,

Thank you for giving us the opportunity to revise and resubmit our manuscript. We greatly appreciate the insightful comments and constructive feedback provided by you and the reviewers, which have been invaluable in helping us strengthen the quality of our work.

We have carefully addressed each of the points raised, and our responses are detailed in the attached “Response to Reviewers” document. We have also included both a marked-up version of our manuscript, highlighting all revisions, and a clean, unmarked version without tracked changes. These updated documents reflect our commitment to ensuring that our manuscript meets the standards and expectations of PLOS ONE.

Should you have any questions or require additional information, please do not hesitate to contact us. We sincerely thank you once again for your time, guidance, and consideration, and we look forward to the opportunity to further improve and clarify our research.

Yours sincerely,

Wenjia Liu, PhD

School of architecture, Zhengzhou University, No. 100 Science Avenue, Zhongyuan Distric

450001 Zhengzhou, China

+86 13523007270

Response to Reviewer #1

Dear Reviewer #1,

Thank you for your thorough review and thoughtful comments on our manuscript. Your feedback has greatly helped us improve both the quality of our work and our own understanding of spatial humanities research. Below is a summary of how we addressed each of your suggestions:

1.Expanded critical discussion

We have revised the Discussion section to include a more in-depth analysis of the three observed spatial morphological features, comparing our findings to other historical Chinese cities. By discussing alternative interpretations and potential explanations for the patterns, we believe our argumentation is now more nuanced and comprehensive.

2.Addressed limitations and data biases

In response to your recommendation, we added a new sub-section titled Limitations of Data Sources in the revised Discussion section. There, we provide a candid assessment of the inherent biases and potential shortcomings in the data, clarifying how these issues might affect the study’s outcomes and conclusions.

3.Elaborated on “why” and “how”

We have expanded our interpretation of the underlying reasons behind the three morphological features by incorporating historical, socio-political, and cultural contexts. This additional discussion explains the broader significance of these spatial patterns and how they may have emerged over time, providing more insight into the “why” and “how” aspects of our results.

4.Strengthened the conclusion

We have revised the Conclusion to offer a more comprehensive overview of our findings, highlighting key contributions and suggesting avenues for future research. We also discuss potential enhancements to data collection and analytical methods that could further refine our approach moving forward.

We sincerely appreciate your meticulous review and believe that your feedback has substantially strengthened our manuscript. Thank you once again for dedicating your time and expertise to improve our work.

Yours sincerely,

Wenjia Liu, PhD

School of architecture, Zhengzhou University, No. 100 Science Avenue, Zhongyuan Distric

450001 Zhengzhou, China

+86 13523007270

Response to Reviewer #2

Dear Reviewer #2,

Thank you for your thorough review and detailed comments on our manuscript. We are truly grateful for the time and effort you spent providing such valuable feedback. Your suggestions have greatly improved the quality of our paper.

Below is a summary of how we have addressed each of your points:

1.Clarification on the “new digital framework”

We acknowledge that our original claim regarding a “new digital framework” may have been overstated. In the revised version, we have removed such references and clarified that, while our application may offer fresh insights into Northern Wei Luoyang and other historical Chinese cities, the core methods (GIS, network analysis) themselves are well-established rather than wholly novel.

2.Enhancements to the Introduction and Literature Review

We have optimized the Introduction section by including more recent literature relevant to the spatial humanities. We also expanded our discussion of the historical significance of Northern Wei Luoyang, thereby placing our study in a broader scholarly context.

3.Data Availability

In the revised manuscript, we provide detailed information on the data we used in GIS and Complex Network analyses, including tables, shapefiles, and Gephi data. We have also deposited these materials in protocols.io for public access, which can be found at:

https://www.protocols.io/blind/D9EBB57CD04511EFA3430A58A9FEAC02

The data has not been officially published yet, but a DOI will be provided after it is finalized.

4.Spatial Analysis Parameters

At your recommendation, we have added detailed parameters for kernel density analysis, mean centers, and standard deviation ellipses. These details appear in the Supporting Information section(S_5 File) and should provide sufficient information for replicating our analyses.

5.Software and Licensing

We confirm that ArcGIS is not freely available, but MARKUS and Gephi are open-source and free to use.

6.Minor Edits and Typographical Corrections

We greatly appreciate your careful reading and have made the necessary corrections based on your suggestions.

We hope these revisions meet your expectations and further enhance the clarity and rigor of our work. Should you have any additional comments or recommendations, we would be most appreciative to receive them.

Yours sincerely,

Wenjia Liu, PhD

School of architecture, Zhengzhou University, No. 100 Science Avenue, Zhongyuan Distric

450001 Zhengzhou, China

+86 13523007270

---

## [Decision Letter · Decision Letter 1]

14 Feb 2025

Spatial Layout Characteristics of Northern Wei Luoyang: A Spatial Humanities Analysis of the “Record of the Monasteries of Luoyang”

PONE-D-24-46751R1

Dear Dr. Liu,

We’re pleased to inform you that your manuscript has been judged scientifically suitable for publication and will be formally accepted for publication once it meets all outstanding technical requirements.

Kind regards,

Yuan Zhang, PhD

Academic Editor

PLOS ONE

Additional Editor Comments (optional):

Reviewers' comments:

Reviewer's Responses to Questions

**Comments to the Author**

1. If the authors have adequately addressed your comments raised in a previous round of review and you feel that this manuscript is now acceptable for publication, you may indicate that here to bypass the “Comments to the Author” section, enter your conflict of interest statement in the “Confidential to Editor” section, and submit your "Accept" recommendation.

Reviewer #1: All comments have been addressed

2. Is the manuscript technically sound, and do the data support the conclusions?

Reviewer #1: Yes

3. Has the statistical analysis been performed appropriately and rigorously? 

Reviewer #1: N/A

4. Have the authors made all data underlying the findings in their manuscript fully available?

Reviewer #1: Yes

5. Is the manuscript presented in an intelligible fashion and written in standard English?

Reviewer #1: Yes

6. Review Comments to the Author

Reviewer #1: (No Response)

7. PLOS authors have the option to publish the peer review history of their article (what does this mean? ). If published, this will include your full peer review and any attached files.

**Do you want your identity to be public for this peer review?** For information about this choice, including consent withdrawal, please see our Privacy Policy .

Reviewer #1: No

---

## [Editor Report · Acceptance letter]

PONE-D-24-46751R1

PLOS ONE

Dear Dr. Liu,

I'm pleased to inform you that your manuscript has been deemed suitable for publication in PLOS ONE. Congratulations! Your manuscript is now being handed over to our production team.

Kind regards,

on behalf of

Professor Yuan Zhang

Academic Editor

PLOS ONE